# ADAPTING VISION-LANGUAGE MODELS FOR EVALUATING WORLD MODELS

## ABSTRACT

World models – generative models that simulate environment dynamics conditioned on past observations and actions – are increasingly central to planning, simulation, and embodied AI. However, evaluating their rollouts remains challenging: existing metrics provide coarse, semantics-agnostic signals, while human evaluation is costly and hard to scale. Effective evaluation requires fine-grained, temporally grounded assessment of action alignment and semantic consistency – capabilities that vision-language models (VLMs) possess but have not been systematically explored for this purpose. We present a case study investigating whether lightweight VLM adaptation can provide reliable semantic evaluation of world model rollouts. We introduce a semantic evaluation protocol targeting two core recognition tasks – action recognition and character recognition – assessed across binary, multiple-choice, and open-ended question formats. To support this protocol, we develop UNIVERSE (*UNI*fied *V*ision-language *E*valuator for *R*ollouts in *S*imulated *E*nvironments), a parameter-efficient VLM adaptation method tailored to rollout evaluation. Through extensive experiments totaling over 5,154 GPU-days, we explore full, partial, and parameter-efficient adaptation methods across various task formats, context lengths, sampling methods, and data compositions. We demonstrate that UNIVERSE matches the performance of task-specific checkpoints while using significantly less training data and parameters. The results demonstrate that VLMs can serve as lightweight, semantics-aware evaluators of world models, and highlight promising directions for extending such evaluators to more complex environments.

## 1 INTRODUCTION

World models – generative models trained to predict future observations conditioned on past observations and actions (Ha & Schmidhuber, 2018; Hafner et al., 2025; Alonso et al., 2024) – are rapidly becoming central to interactive AI. They provide a powerful abstraction for learning, reasoning, and planning in complex interactive environments, and underpin advances in neural game engines (Kanervisto et al., 2025; Guo et al., 2025; Gao et al., 2025; Chen et al., 2025), embodied AI (Du et al.; Yang et al., 2024), and autonomous driving (Russell et al., 2025; Hu et al., 2023a; Ni et al., 2025).

Yet, evaluating world models remains a bottleneck. Rollouts are semantically rich and temporally grounded, requiring metrics that assess (i) alignment between generated frames and action sequences at the timestamp level (Yang et al., 2024), and (ii) consistent entity tracking over time (Kanervisto et al., 2025). Existing approaches fall short: (i) early distributional metrics focus on images and are sensitive to low-level variations (Salimans et al., 2016; Heusel et al., 2017; Binkowski et al., 2018), (ii) motion-aware metrics like FVD (Unterthiner et al., 2018) lack semantic grounding, and (iii) multimodal metrics ignore timestamp-level action conditioning (Jayasumana et al., 2024). Emerging text-to-video benchmarks (Liu et al., 2024b; Huang et al., 2024; Liao et al., 2024) focus on open-ended generation but neglect the fine-grained control central to world model evaluation. Even cutting-edge LLMs fail in this setting (Appendix G.1, Figure 15). Human evaluation remains the gold standard (Agarwal et al., 2025; Analysis, 2024), however, it remains costly and hard to scale.

To address this gap, we propose a novel evaluation protocol targeting two important dimensions of rollout quality: action alignment and character consistency, formalized as recognition tasks – Action Recognition (AR), Character Recognition (CR) – across formats of varying complexity. The protocol

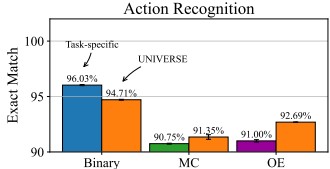 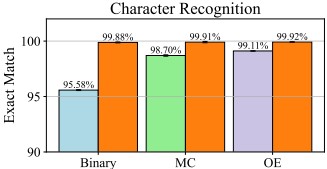 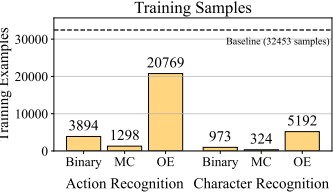

Figure 1: Performance and efficiency of UNIVERSE (orange bars throughout) compared to task-specific baselines (multiple colours), all models trained for 10 epochs. **Left and Center:** Action recognition and Character Recognition accuracy across binary, multiple-choice, and open-ended settings. **Right:** Sample efficiency – our adaptation recipe achieves strong performance with substantially fewer training samples per epoch.

provides a foundation for semantic evaluation of rollouts, and extends upon existing evaluation method by providing insight into semantic quality of generated rollouts. Inspired by the LLM-as-a-Judge research direction Zheng et al. (2023), we explore whether Vision-Language Models (VLMs) can serve as effective evaluators in this setting. VLMs have demonstrated strong generalization across multimodal tasks (Li et al., 2023; Driess et al., 2023; Chen et al., 2023; Wang et al., 2024; Abdin et al., 2024; Liu et al., 2024a; Deitke et al., 2024; McKinzie et al., 2024), and show promise as automatic judges of generative models (Lee et al., 2024; Mañas et al., 2024; Lin et al., 2024; Chen et al., 2024). Yet, off-the-shelf VLMs struggle with the temporal grounding and domain-specific knowledge (see Sec. 4, Zero-Shot Evaluation). Limited resources and sparse text supervision introduce another layer of complexity to the setting.

We therefore conduct a systematic study of adaptation strategies under realistic data and compute constraints. Throughout the study totaling over 5,154 GPU-days, we analyze the impact of supervision regime, frame sampling strategy, visual context length, and training budget. The result of the study is UNIVERSE (*UNI*fied *V*ision-language *E*valuator for *R*ollouts in *S*imulated *E*nvironments), a lightweight, adaptable, and semantics-aware evaluator for world models. UNIVERSE achieves parity with six task-specific checkpoints while using a single unified model (see Figure 1. To validate reliability, we conduct a large-scale human study on rollouts spanning diverse environments, model scales, and rollout fidelities. UNIVERSE's judgments show strong agreement with human ratings, demonstrating its effectiveness as a practical, semantics-aware evaluator—particularly valuable in settings where ground-truth annotations are unavailable or prohibitively expensive. To support reproducibility, we release our code, evaluation dataset, and human annotation dataset[1].

## 2 RELATED WORK

**Challenges in Evaluating World Models.** World models are generative systems that learn predictive representations of environment dynamics (Ha & Schmidhuber, 2018), originally proposed for model-based RL (Sutton, 1991) and now central to domains such as neural game engines (Kanervisto et al., 2025; Guo et al., 2025; Gao et al., 2025; Chen et al., 2025), embodied AI (Du et al.; Yang et al., 2024), and autonomous driving (Russell et al., 2025; Hu et al., 2023a; Ni et al., 2025). Recent models such as Dreamer v1–3 (Hafner et al., 2020; Hafner et al.; 2023; 2025), MuZero (Schrittwieser et al., 2020), IRIS (Micheli et al., 2023), UniSim (Yang et al., 2024), and DIAMOND (Alonso et al., 2024) have improved rollout fidelity and controllability. Yet evaluation largely focuses on downstream success metrics—e.g., game score or goal completion (Bellemare et al., 2013; Kaiser et al., 2020; Guss et al., 2021; Baker et al., 2022; Beattie et al., 2016)—which provide only coarse, indirect signals of rollout quality. Genie (Bruce et al., 2024; Parker-Holder et al., 2024) decouples world model learning and agent training, but its evaluation still emphasizes visual quality and control, without probing semantic or causal fidelity. Cosmos (Agarwal et al., 2025) proposes a structured protocol that combines FID/FVD with structure-from-motion-based 3D consistency checks and human ratings on instruction following, object permanence, and visual verity. While insightful, this approach is tied to simulator-specific infrastructure and requires costly manual comparison. Human-in-the-loop

---

[1] https://anonymous.4open.science/r/vlms-for-wms-2651/README.md

protocols such as the Video Generation Arena (Analysis, 2024) also rely on pairwise comparison to assess rollout quality. These methods, though informative, are expensive and hard to scale.

**Evaluation Metrics and Protocols for Visual Generation.** Early evaluations of generative models relied on full-reference metrics such as PSNR and SSIM (Wang et al., 2004), which capture pixel-level and perceptual similarity but are sensitive to spatial misalignments and fail to reflect semantic fidelity. To address this, distributional metrics like Inception Score (IS) (Salimans et al., 2016), Fréchet Inception Distance (FID) (Heusel et al., 2017), and Kernel Inception Distance (KID) (Binkowski et al., 2018). Other proposals such as PPL (Karras et al., 2019), Parzen likelihoods (Goodfellow et al., 2014), and HYPE (Zhou et al., 2019) attempt to quantify perceptual smoothness or human realism, but remain focused on static images. For video generation, FVD (Unterthiner et al., 2018) generalizes FID using I3D features (Carreira & Zisserman, 2017), introducing a motion-aware distributional baseline. Yet, FVD also lacks semantic grounding and does not account for causal structure or goal alignment. To improve semantic grounding, metrics based on text-image alignment have been proposed. CLIPScore (Hessel et al., 2021) and CLIPSIM (Wu et al., 2021) compute similarity between generated visuals and textual or visual references using CLIP embeddings (Radford et al., 2021), while Jayasumana et al. (2024) extend this to distributional comparisons via MMD. However, all operate at the frame level. Structured evaluation protocols using vision-language reasoning have also emerged. VQA Accuracy (Mañas et al., 2024) uses LLMs to score answers on static image questions, and VQAScore (Lin et al., 2024) probes alignment via templated binary queries. Lee et al. (2024) propose VLM evaluator to evaluate other VLMs responses given user criteria. These approaches introduce task structure but remain limited to single-frame evaluation. Recent text-to-video (T2V) benchmarks such as EvalCrafter (Liu et al., 2024b), VBench (Huang et al., 2024), and DEVIL (Liao et al., 2024) introduce curated prompts and metrics covering text alignment, motion realism, and perceptual quality. While these protocols push forward evaluation of open-ended video generation, they lack timestamp-level action grounding.

**Vision-Language Model Adaptation.** VLMs have emerged as powerful tools for multimodal understanding, demonstrating strong performance across tasks such as captioning, retrieval, visual question answering, and instruction following (Hendriksen et al., 2022; Li et al., 2023; Driess et al., 2023; Chen et al., 2023; Wang et al., 2024; Abdin et al., 2024; Liu et al., 2024a; Deitke et al., 2024; McKinzie et al., 2024). Adaptation approaches can be broadly categorized into prompt-level and weight-level methods. One prominent prompt-level adaptation techniques is prompt tuning, which injects task information directly into the input space (Miyai et al., 2023; Zhou et al., 2024; Wu et al., 2024a), and in-context learning (ICL), where models such as GPT-3 (Brown et al., 2020) and Flamingo (Alayrac et al., 2022) condition on task demonstrations at inference time without updating parameters. Retrieval-augmented generation (RAG) (Lewis et al., 2020) combines parametric models with non-parametric memory, and multimodal variants incorporate external visual or auditory context (Hu et al., 2023b; Chen et al., 2022a). While lightweight, these approaches are limited in their ability to model temporal dependencies or align with structured rollouts. Weight-level adaptation enables stronger domain alignment but incurs higher computational cost. Full finetuning remains effective yet costly, while partial finetuning (Ye et al., 2023) offers a trade-off by updating only selected layers. Parameter-efficient finetuning (PEFT) provides a scalable alternative and can be grouped into low-rank and adapter-based strategies (Han et al., 2024). Low-rank methods, such as LoRA (Hu et al., 2022), inject rank-constrained updates into frozen layers. Recent extensions improve upon this via weight decomposition (Liu et al.), quantization-aware adaptation (Dettmers et al., 2023; Xu et al., 2024), mixture-of-experts routing (Wu et al., 2024b), and long-context support (Chen et al.). Adapter-based methods insert lightweight modules between frozen layers to enable modular adaptation with minimal overhead (Luo et al., 2023; Zhao et al., 2024). A parallel line of work investigates multimodal few-shot learning. Frozen (Tsimpoukelli et al., 2021) was among the first to explore this setting, followed by works combining prompting and ICL for improved sample efficiency (Jin et al., 2022; Song et al., 2022), and works introducing a learnable meta-mapper to bridge frozen VLM components for few-shot meta-learning (Najdenkoska et al., 2023).

**Our Focus.** While prior efforts have explored related challenges, none directly address the evaluation of the structured, action-conditioned fidelity and semantics of world model rollouts using adapted VLM. To this end, we introduce: (i) an evaluation protocol for world model rollouts, targeting fine-grained, temporally grounded assessment of semantic fidelity; (ii) UNIVERSE, a VLM-based method to support the protocol. We validate its alignment with human judgments and demonstrate its scalability and semantic sensitivity across rollout conditions.

## 3 METHODOLOGY

We consider the problem of evaluating rollouts generated by *world models* in interactive environments. A world model $W$ is trained to predict the next observation $o_t$ given the past observations $o_{<t}$ and actions $a_{<t}$: $W : (o_{<t}, a_{<t}) \rightarrow o_t$, where $o_t \in \mathcal{O}$ represents the sensory observation at timestep $t$, typically an RGB image. Rollouts consist of temporally grounded sequences that reflect the causal effects of control inputs. These outputs are semantically rich and visually complex, requiring timestamp-level assessment of correctness.

To enable automatic evaluation, we propose UNIVERSE, an adapted Vision-Language Model (VLM) that serves as a structured evaluator for world model rollouts. Formally, it operates as a function: $E : (V, Q) \rightarrow \hat{A}$, where $V = (o_{t_1}, \ldots, o_{t_k}) \in \mathcal{O}^k$ is a sequence of frames from a rollout, $Q \in \mathcal{L}$ is a natural language question, and $\hat{A} \in \mathcal{L}$ is the predicted answer. Evaluation quality is measured by comparing $\hat{A}$ to the reference answer $A$ using semantic similarity metrics.

**Evaluation Protocol.** We define two structured recognition tasks: (i) *Action Recognition (AR)*: Assesses whether generated sequences accurately reflect the effects of agent actions given the segment; (ii) *Character Recognition (CR)*: Evaluates whether entities maintain consistent identity and appearance across time. Each task is framed as a visual QA problem: the evaluator receives a sequence of frames and a natural language prompt (binary, multiple-choice, or open-ended), and generates a textual response. Outputs are scored using Exact Match (EM) and ROUGE-$F_1$ (ROUGE), capturing both literal and semantic alignment with the reference answer. Metric details are in Appendix E.2.

**Dataset Construction.** Effective VLM adaptation for rollout evaluation requires a dataset that (i) captures realistic human behavior in interactive environments, and (ii) aligns with prior work in simulated settings to support comparability and reproducibility. To satisfy these constraints, we partnered with Ninja Theory and curated a dataset from both internal and public *Bleeding Edge* gameplay recordings, focusing on the *Skygarden* evnironment (Kanervisto et al., 2025). This dataset provides high visual and behavioral diversity (Pearce et al., 2025), includes a publicly available evaluation split, and is closely aligned with prior work in the domain (Kanervisto et al., 2025; Pearce et al., 2025; Tot et al., 2025; Sharma et al., 2024; Devlin et al., 2021), enabling fair comparison.

Data preparation proceeds in three stages: (i) *Preprocessing:* Segment gameplay into 14-frame clips with synchronized video, control logs, and metadata; (ii) *Description Generation:* Convert structured annotations (e.g., actions, agent states) into natural language summaries; (iii) *Question-Answer Pair Construction:* Generate six complimentary QA pairs per clip (binary, multiple-choice, and open-ended) spanning both AR and CR tasks. The final dataset contains 32.453 training clips and 8.113 validation clips, yielding 194.718 and 48.678 QA pairs, respectively. See Appendix D for details.

**Model Architecture.** We adapt a model from the PaliGemma family (Beyer et al., 2024; Steiner et al., 2024), consisting of a vision encoder $\mathcal{M}_V$, a projection head $\mathcal{M}_P$, and a language decoder $\mathcal{M}_L$. Based on initial zero-shot evaluations (Appendix G.2), we use a single configuration for all experiments—PaliGemma 2 3b, which includes a 2B-parameter Gemma 2 decoder pretrained on 2T tokens. Input frames are resized to $224 \times 224$ and tokenized into 256 patches each. Model achitecture details are in Appendix E.1.

Each model input sequence $S = \{S_\mathcal{I}, S_\mathcal{T}^{\text{PREF}}, S_\mathcal{T}^{\text{SUFF}}\}$ consists of: visual tokens $S_\mathcal{I}$ from $k$ frames, a textual prefix $S_\mathcal{T}^{\text{PREF}}$ containing the task-language cue and question, and a suffix $S_\mathcal{T}^{\text{SUFF}}$ with the expected answer (used only during training). This format allows the decoder to attend jointly over visual and textual context. Full prompt details are provided in Appendix E.1.

**Training Objective.** We optimize a causal language modeling loss on the answer suffix:

$$\mathcal{L}(S) = - \sum_{t=1}^{T_{\text{SUFF}}} \log P(s_t^{\text{SUFF}} \mid S_{<t'}) \tag{1}$$

where $s_t^{\text{SUFF}}$ is the $t$-th token in the suffix, and $t' = T_\mathcal{I} + T_{\text{PREF}} + t$ is the token position in the flattened sequence.

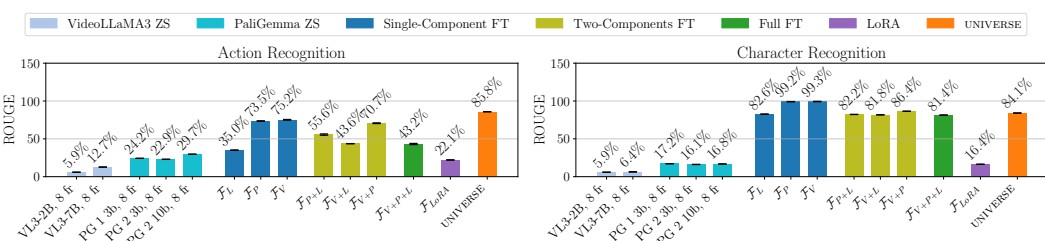

Figure 2: Comparison of UNIVERSE and baseline models on Action and Character Recognition, all models trained for 1 epoch. **Left**: UNIVERSE outperforms all baselines on AR. **Right**: On CR, it ranks third, behind models with either full vision encoder tuning or task-specific training with greater supervision. Trained under a unified protocol with minimal parameter updates (0.07%) and reduced per-task data, UNIVERSE delivers strong performance across both tasks.

**Adaptation Strategies.** We explore a broad design space for adapting pretrained VLMs to temporally grounded rollout evaluation. Our study spans three core dimensions: *fine-tuning configurations*, *frame sampling policy*, and *supervision composition*.

*Fine-Tuning Configurations.* We compare five adaptation strategies varying in parameter count and modularity: (i) *Zero-shot prompting:* No tuning; model is prompted directly. (ii) *Full fine-tuning:* All parameters $\theta = \theta_V \cup \theta_P \cup \theta_L$ are updated end-to-end. (iii) *Dual-component fine-tuning:* Two of three modules are trained (e.g., $\theta_P \cup \theta_L$). (iv) *Single-component fine-tuning:* Only one module—vision, projection, or language—is updated. (v) *Parameter-efficient fine-tuning:* We apply LoRA (Hu et al., 2022) adapters to attention and MLP layers in vision and language components: $\mathbf{W} \leftarrow \mathbf{W} + \frac{\alpha}{r}\mathbf{AB}$, $\alpha = 8$, $r \in \{8, 16, 32, 48, 64\}$.

*Frame Sampling Policy.* We vary both the number of input frames and their sampling strategy. Specifically, we sweep over $k \in [1, 8]$, and evaluate two selection methods: (i) *First-n*: selecting the first $k$ frames from each rollout; (ii) *Uniform-n*: sampling $k$ frames uniformly across the full clip.

*Supervision Composition.* To support generalization across QA formats and tasks, we construct a multi-task dataset covering binary, multiple-choice, and open-ended prompts across both AR and CR. We perform a three-stage grid search to optimize the data mixture: (i) Varying AR/CR task ratios ($\alpha_{\text{AR}}$, $\alpha_{\text{CR}}$) while fixing QA type proportions ($\beta_{\text{Binary}}$, $\beta_{\text{MC}}$, $\beta_{\text{OE}}$); (ii) Tuning the proportion of open-ended supervision ($\beta_{\text{OE}}$) for best performance; (iii) Adjusting $\beta_{\text{Binary}}$ and $\beta_{\text{MC}}$.

**UNIVERSE: UNIfied Vision-language Evaluator for Rollouts in Simulated Environments.** We distill our empirical findings into UNIVERSE, a lightweight and scalable adaptation method for temporally grounded evaluation of world model rollouts using VLMs. Designed for constrained compute and limited supervision, UNIVERSE delivers strong generalization across our evaluation protocol using a single, partially tuned model. The method combines three main components: I *Partial fine-tuning:* We update only the projection head ($\theta_P$), training just 0.07% of model parameters. Despite this minimal footprint, it achieves the second-best performance among all strategies—trailing only vision encoder tuning, which requires ∼11% of parameters and incurs significantly higher compute cost. II *Efficient frame sampling:* Each input sequence includes $k = 8$ frames sampled uniformly from a 14-frame rollout. This sparsity-aware strategy maintains long-range temporal structure while reducing token count and enabling efficient batching. III *Mixed supervision:* We train on a hierarchical mixture of tasks and QA formats. The task distribution favors Action Recognition ($\alpha_{\text{AR}} = 0.8$) as it converges slower. Within each task, we emphasize open-ended questions ($\beta_{\text{OE}} = 0.8$), while maintaining smaller proportions of binary ($\beta_{\text{binary}} = 0.15$) and multiple-choice ($\beta_{\text{MC}} = 0.05$) examples.

## 4 EXPERIMENTS

We evaluate UNIVERSE on ground truth video data, focusing on AR and CR across binary, multiple-choice, and open-ended formats. Our goals are twofold: (i) to benchmark performance against zero-shot and fine-tuned baselines, and (ii) to assess the trade-offs between adaptation strategies under constrained supervision and compute.

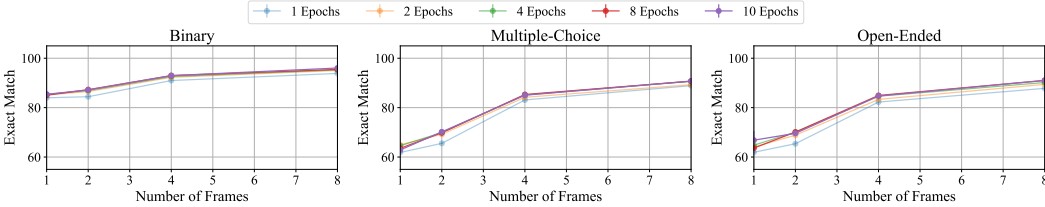

Figure 3: Action Recognition performance as a function of training supervision (epochs) and temporal context (number of frames), evaluated across all formats. Performance improves along both axes, with highest accuracy achieved when both dimensions are scaled.

**Baselines.** We compare UNIVERSE against two classes of baselines: (i) *Zero-shot VLMs:* Seven off-the-shelf models, including VideoLLaMA3 (2B, 7B) (Boqiang Zhang, 2025) and three PaliGemma models: version 1 (3B) and version 2 (3B and 10B) (Beyer et al., 2024; Steiner et al., 2024), evaluated without domain adaptation using an 8-frame visual context window.[2] (ii) *Fine-tuned PaliGemma 2:* Variants adapted via full, partial, and parameter-efficient tuning. This backbone is selected based on a sweep over PaliGemma variants, using zero-shot performance as a guide (Appendix G.2). The adaptation space includes 8 primary baselines: (i) *Single-component fine-tuning*: tuning only the vision encoder ($\mathcal{F}_V$), the multimodal projector ($\mathcal{F}_P$), or the language head ($\mathcal{F}_L$); (ii) *Two-component fine-tuning*: jointly tuning pairs of components—$\mathcal{F}_{V+P}$, $\mathcal{F}_{V+L}$, and $\mathcal{F}_{P+L}$; (iii) *Full-model fine-tuning*: tuning all components simultaneously ($\mathcal{F}_{V+P+L}$); (iv) *LoRA-based tuning*: Parameter-efficient adaptation with rank $r = 8$, selected after observing minimal performance variation across $r \in \{8, 16, 32, 48, 64\}$ (see Appendix G.4 for details). All models are trained using 8-frame clips and a single epoch.

**Results.** Figure 2 (left, center) summarizes performance across Action Recognition (AR) and Character Recognition (CR). Zero-shot VLMs perform poorly: VideoLLaMA3 variants stay below 12.7% (AR) and 6.4% (CR), while PaliGemma reaches 29.7% (AR) and 17.2% (CR), confirming that general-purpose models lack the temporal grounding and domain-specific semantics needed for rollout evaluation. In contrast, UNIVERSE outperforms all models on AR and ranks third on CR—despite tuning only the 2.66M-parameter projector (0.07% of the model) under a unified protocol spanning both tasks, all prompt formats, and reduced supervision. The two stronger CR baselines fine-tune either the full 400M-parameter vision encoder or use $5\times$ more task-specific CR data. Its performance under these constraints underscores the efficiency and generality of our adaptation strategy for temporally grounded evaluation.

## 5 ANALYSIS

We find a consistent performance gap between AR and CR, highlighting the greater temporal and causal complexity of action understanding. This motivates our focus on AR as the more challenging and diagnostic task. Below, we analyze how adaptation choices shape performance on AR.

**Supervision and Temporal Context.** We begin by analyzing how supervision (training budget) and temporal input (number of frames) influence UNIVERSE performance. By independently and jointly varying the number of training epochs and input frames, we disentangle the contributions of model capacity and temporal context to task success.

*Results.* CR converges rapidly, achieving over 97% exact match after 12.5% of an epoch (~4K samples; Figure 5, bottom), and shows minor improvement with further training. In contrast, AR improves only modestly under extended training when limited to a single frame (Figure 5, top), suggesting that supervision alone is insufficient in the absence of temporal information. Motivated by this, we jointly scale both supervision and input length, varying the number of frames and epochs. As shown in Figure 3, performance on AR improves consistently across all formats, with the best results achieved under combined scaling.

---

[2]We also experimented with CLIPScore-based evaluation (Appendix G.3); results underperformed relative to selected baselines and were constrained to predefined candidate sets, further underscoring the need for model adaptation.

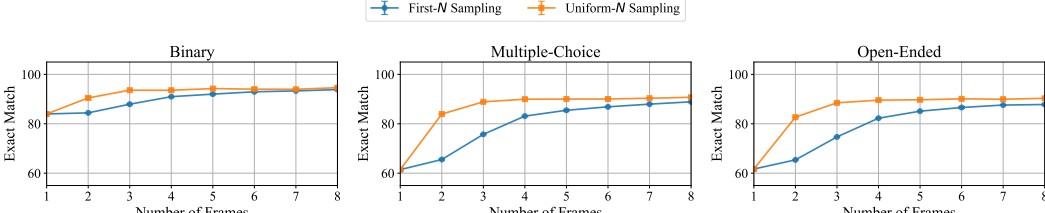

Figure 4: Effect of frame sampling strategy on Action Recognition performance across all formats. Uniform-$n$ sampling (orange) consistently outperforms first-$n$ (blue), with especially large gains at low frame counts, and maintains an advantage as temporal context increases.

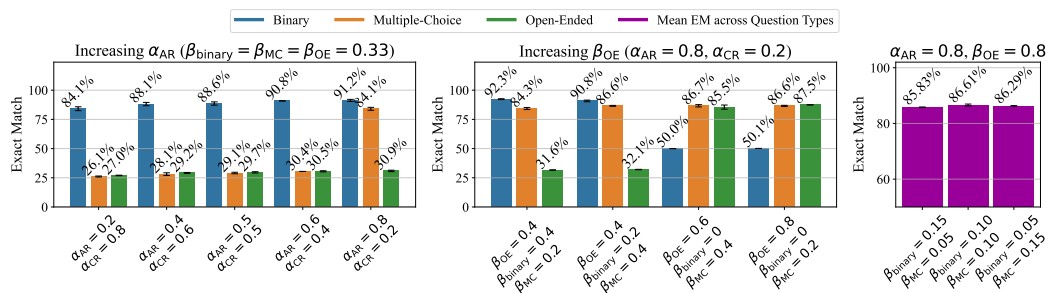

Figure 6: Hierarchical ablation of training data composition for UNIVERSE. **Left**: Varying task-level ratio $\alpha$ (AR vs. CR) with uniform format distribution ($\beta = 1/3$) shows that increasing $\alpha_{AR}$ improves AR performance, especially for multiple-choice, while open-ended remains flat. **Center**: Sweeping format-level ratio $\beta_{OE}$ with fixed $\alpha_{AR} = 0.8$ reveals that oversampling open-ended data ($\beta_{OE} = 0.8$) improves AR-OE performance. **Right**: Fine-tuning binary and MC proportions under $\beta_{OE} = 0.8$ shows performance is stable across mixes, with slight gains from $\beta_{binary} = 0.15$, $\beta_{MC} = 0.05$.

**Temporal Sampling Strategies.** Following the observation that AR requires both extended supervision and temporally rich input, we examine how frame selection impacts performance. We compare first-$n$ sampling, which selects the first $n$ consecutive frames from each rollout, to uniform-$n$ sampling, which draws $n$ evenly spaced frames across the entire sequence. We conduct experiments at varying context lengths, using $n \in \{1, 2, \ldots, 8\}$ frames. to evaluate the impact of both sampling method and input horizon.

*Results.* As shown in Figure 4, uniform-$n$ consistently outperforms first-$n$ across all evaluation formats. The effect is most pronounced at low frame counts. With only 2 input frames, uniform sampling improves exact match accuracy from 84.42% to 90.47% in Binary, from 65.53% to 83.93% in Multiple-Choice, and from 65.38% to 82.68% in Open-Ended formats. Gains persist even at 8 frames, where uniform sampling maintains an advantage across formats.

**Optimizing Data Mix for Unified Multi-Task Evaluation.** We analyze how training data composition affects multi-task performance in UNIVERSE, with the goal of enabling a single model to generalize across AR and CR. Specifically, we study how the task-level ratio $\alpha$ and format-level ratio $\beta$ influence performance across evaluation settings. We first conduct a hierarchical ablation to identify an optimized data mixture, then assess its impact by comparing against a default task mix with uniform sampling.

*Data Mix Optimization.* To determine an effective training mixture for UNIVERSE, we perform a hierarchical ablation over task-level and format-level data ratios. We begin by varying the task-level proportion $\alpha$ (AR vs. CR), holding the format distribution fixed at

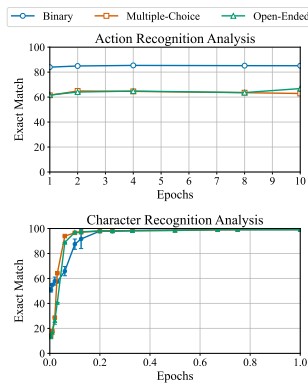

Figure 5: Exact Match accuracy for Action Recognition and Character Recognition.

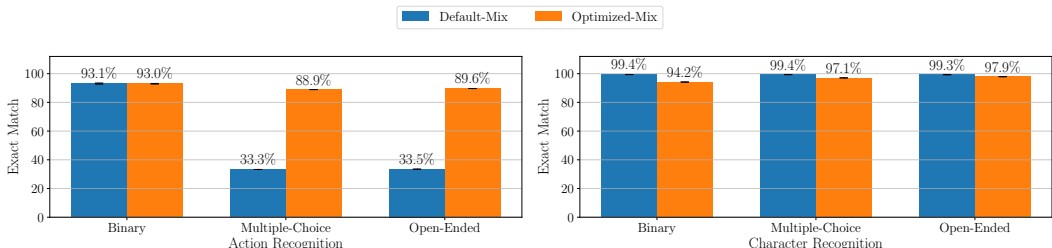

Figure 7: Comparison of two training regimes: a default data mix (equal task and format proportions) and the optimized mix derived from hierarchical tuning. The optimized configuration yields substantial gains on AR, while maintaining strong CR performance.

$\beta_{\mathtt{binary}} = \beta_{\mathtt{MC}} = \beta_{\mathtt{OE}} = \frac{1}{3}$. As shown in Figure 6 (left), increasing $\alpha_{\mathtt{AR}}$ improves AR performance—especially for multiple-choice—while CR remains stable, with a favorable tradeoff reached at $\alpha_{\mathtt{AR}} = 0.8$. However, open-ended accuracy shows little change, motivating format-specific rebalancing. Fixing $\alpha_{\mathtt{AR}} = 0.8$, we sweep the format ratio $\beta_{\mathtt{OE}}$, and observe in Figure 6 (center) that AR-OE accuracy improves substantially with increased open-ended coverage, peaking at $\beta_{\mathtt{OE}} = 0.8$, albeit at the cost of binary performance. To restore balance, we fix $\beta_{\mathtt{OE}} = 0.8$ and allocate the remaining budget across binary and multiple-choice formats. As shown in Figure 6 (right), performance remains robust across configurations, with a slight preference for $\beta_{\mathtt{binary}} = 0.15$ and $\beta_{\mathtt{MC}} = 0.05$. Based on these findings, we adopt the following optimized data composition: $\alpha_{\mathtt{AR}} = 0.8$, $\alpha_{\mathtt{CR}} = 0.2$; $\beta_{\mathtt{binary}} = 0.15$, $\beta_{\mathtt{MC}} = 0.05$, and $\beta_{\mathtt{OE}} = 0.8$.

*Effectiveness of the Optimized Mix.* Having identified an optimized training mixture through hierarchical ablation, we now evaluate its impact in practice. We compare the final UNIVERSE model—trained with this optimized mix—to a baseline trained with a default task and format distribution. We train both models on 4 epochs. As shown in Figure 7, the optimized configuration yields substantial gains on AR, particularly for multiple-choice and open-ended formats, while maintaining competitive performance on CR. These results underscore the importance of data composition in enabling robust multi-task learning within a unified evaluator.

# 6 EVALUATING WORLD MODEL ROLLOUTS WITH UNIVERSE

We evaluate the reliability of UNIVERSE through a human study spanning eight distinct settings that vary in model scale, training data diversity, and output resolution. Our analysis considers two axes: (i) *in-domain accuracy*, measured on Skygarden, and (ii) *generalization* to six previously unseen environments.

Concretely, we study rollouts generated by: (i) a large-scale model trained across multiple environments with higher-resolution rollouts ($300 \times 180$), and (ii) a smaller model trained on a single environment with lower-resolution rollouts ($128 \times 128$). These two model families expose complementary challenges: resolution mismatch, domain coverage, and rollout fidelity. We construct eight evaluation settings: settings 1–7 draw from the large-scale model across diverse environments, while setting 8 uses the smaller model in the fine-tuning environment. Each setting contains 30 rollouts, paired with six natural-language questions from our evaluation protocol, yielding 240 rollouts in total. UNIVERSE answers each question via majority vote over five greedy decoding samples. Human annotators rate responses on a four-point ordinal scale (*Correct*, *Partially Correct*, *Incorrect*, *Unclear*), with double annotation and adjudication on disagreements. Inter-rater reliability is measured using Cohen's $\kappa$. Full details of the annotation protocol are provided in Appendix F.

*Results.* Figure 9 reports graded accuracy across all settings. Rollouts from the smaller, single-environment model yield lower evaluation accuracy, likely due to resolution mismatch, while the larger, multi-environment model provides higher-quality inputs. These results demonstrate that UNIVERSE generalizes across model scales, rollout fidelities, and environments, while remaining closely aligned with human judgments.

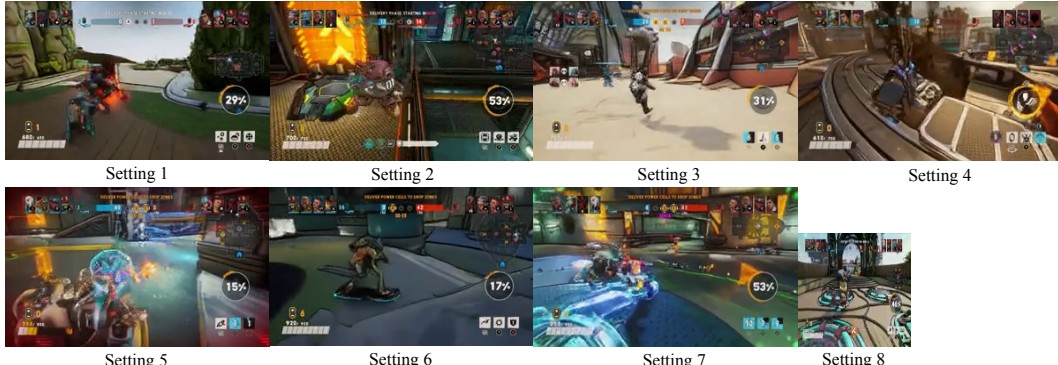

Figure 8: Example frames from the eight evaluation settings, spanning different model scales, rollout fidelities, and environments. Note the resolution difference: Settings 1-7 feature $300 \times 180$ rollouts, whereas Setting 8 uses $128 \times 128$.

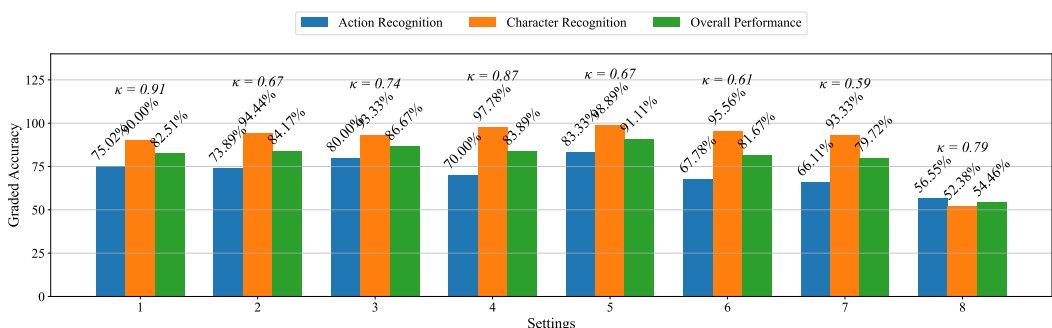

Figure 9: Graded accuracy of UNIVERSE across the eight evaluation settings. Performance improves with higher-fidelity rollouts and remains stable across unseen environments (2–7). Cohen's $\kappa$ indicates substantial inter-rater agreement.

# 7    CONCLUSION

World model evaluation remains a fundamental challenge, requiring fine-grained assessment of semantic consistency and action alignment – capabilities poorly addressed by existing metrics. In this work, we investigate whether lightweight VLM adaptation can provide reliable semantic evaluation of world model rollouts through a comprehensive case study. We introduce a structured evaluation protocol centered on action and character recognition tasks across binary, multiple-choice, and open-ended formats. To support this, we propose UNIVERSE, a unified method for adapting VLMs to this setting through mixed supervision, efficient frame sampling, and lightweight fine-tuning. Our large-scale study demonstrates that UNIVERSE matches the performance of task-specific checkpoints using a single unified model and aligns closely with human judgments, establishing it as a lightweight, semantics-aware evaluator for evaluating world models.

**Limitations.** While our experiments focus on simulated environments, chosen both for their ground-truth availability and their direct relevance to large video-game and interactive-entertainment industries (Kanervisto et al., 2025), evaluating UNIVERSE in real-world settings remains an important next step. Our protocol focuses on foundational semantic tasks, and extending it to cover higher-level reasoning represents an important direction for future work. Scaling UNIVERSE to long-horizon rollouts is also challenging, as fine-grained reasoning becomes harder with larger visual context. Our frame-sampling analysis suggests that more intelligent sampling or hierarchical summarization could address this, and we plan to explore such strategies. Finally, as with all pretrained VLMs, UNIVERSE may inherit biases and exhibit reduced reliability on ambiguous cases (Bleeker et al., 2024). While we examined them partially during evaluation, a deeper investigation into bias propagation is an exciting research direction for future work.

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

# ADAPTING VISION-LANGUAGE MODELS FOR EVALUATING WORLD MODELS

### APPENDIX

### TABLE OF CONTENTS

## A  BROADER IMPACT

As world models become integral to simulation, planning, and decision-making in interactive environments, evaluation remains a key bottleneck for both research progress and safe deployment. We address this challenge by introducing a unified, sample-efficient framework for evaluating world model rollouts using adapted VLMs, designed for fine-grained, temporally grounded, and semantically coherent assessment.

This capability has direct implications for high-impact domains such as neural game engines (Kanervisto et al., 2025; Guo et al., 2025; Gao et al., 2025; Chen et al., 2025), embodied AI (Du et al.; Yang et al., 2024), and autonomous driving (Russell et al., 2025; Hu et al., 2023a; Ni et al., 2025), where world models simulate environment dynamics and support downstream control and generalization. In such contexts, precise and interpretable evaluation is critical not only for benchmarking, but also for diagnosing failure modes and ensuring alignment with intended behaviors.

By reducing dependence on human annotation and task-specific fine-tuning, UNIVERSE offers a scalable alternative that lowers the computational and environmental costs of rollout evaluation. However, reliance on automated evaluators introduces risks: adapted VLMs may inherit biases from pretraining, struggle under distributional shift, or yield unreliable judgments in edge cases. These risks are amplified in safety-critical settings, where miscalibrated evaluations can propagate downstream errors.

We therefore advocate for cautious deployment, accompanied by human oversight, rigorous validation, and transparent reporting. While UNIVERSE advances the automation of world model evaluation, it must be situated within evaluation pipelines that foreground robustness, interpretability, and accountability.

## B  REPRODUCIBILITY STATEMENT

To support reproducibility and facilitate future research, we provide detailed instructions for reproducing all main experiments. Detailed descriptions of model architectures, training procedures, and dataset construction are provided in Section 4 and Appendix E. A high-level overview of the overall implementation framework is included in Appendix C.1. All experiments have been repeated for three runs. Plots and tables with quantitative results show the standard deviation across these runs.

*Use of Existing Assets.*  We experiment with a range of open-weight VLMs, including three PaliGemma variants (version 1 (3B) (Beyer et al., 2024) and version 2 (3B and 10B) (Steiner et al., 2024)), VideoLLaMA3 (2B, 7B) (Boqiang Zhang, 2025), and CLIP (Radford et al., 2021) with the following vision encoder configurations: ViT-B/32, ViT-B/16, ViT-L/14, and ViT-L/14 with $336 \times 336$ resolution. UNIVERSE is built on top of PaliGemma v2 (3B), using publicly released checkpoints for initialization. Further architectural and implementation details are provided in Appendix E.1. For our software stack, we use Matplotlib (Hunter, 2007) for plotting, NumPy (Harris et al., 2020) for data handling, openCV (Bradski, 2000), FFmpeg (Tomar, 2006) and PIL (Umesh, 2012) for video and image processing, and NLTK (Bird & Loper, 2004) for text processing. Parameter-efficient fine-tuning is implemented using the PEFT library (Mangrulkar et al., 2022). We log our experiments using Weights and Biases (Biewald, 2020).

*Use of Large Language Models.* Portions of the manuscript were polished with the assistance of large language models (LLMs). The use of LLMs was limited to only improving readability and style; all ideas, and experimental designs were developed by the authors.

*Compute Resources.* All experiments were conducted using NVIDIA A100 GPUs (40GB memory) on an internal compute cluster. Each model was trained and/or evaluated using 8 GPUs. The compute breakdown is as follows: zero-shot evaluation experiments consumed approximately 136 GPU-days; baseline fine-tuning experiments required around 864 GPU-days; analysis experiments contributed the bulk of usage, totaling 2,554 GPU-days. Human evaluation experiments—including rollout generation and response annotation using UNIVERSE—incurred an additional 1.125 GPU-days. Additional compute was required for preliminary experiments, and failed runs not included in the final paper. These development activities accounted for an estimated 1,599 GPU-days. In total, all experiments amounted to approximately 5,153 GPU-days, equivalent to 14.12 GPU-years.

## C UNIVERSE: ADDITIONAL DETAILS

### C.1 IMPLEMENTATION OVERVIEW

This section outlines the implementation of UNIVERSE in Python, presented as high-level pseudocode. The system is structured around two main stages: (i) *Adaptation:* fine-tuning a VLM on task-specific question-answer (QA) supervision derived from ground truth; (ii) *Evaluation:* using the adapted model to assess new rollouts via structured, prompt-based recognition tasks.

**Adaptation Pipeline.**

The adaptation stage can be implemented as two modules: `AdaptationDatasetBuilder` and `VLMAdapter`.

**AdaptationDatasetBuilder.** This class constructs an adaptation dataset from raw ground truth data, initialized via `load_ground_truth_data` (see Section 3 and Appendix D). The core method, `build`, takes four arguments: `alpha_task`, which specifies the task mixture ratio; `beta_format`, which controls the distribution over QA prompt formats; `context_length`, which determines the number of frames per QA instance; and `sampling_strategy`, which defines how frames are sampled from rollouts. The builder first applies `stratified_sample` to select a subset of annotated samples that match the specified configuration. For each sample, it invokes `_sample_visual_context` to extract the relevant frames, and constructs a triplet consisting of `frames`, `question`, and `answer`.

**VLMAdapter.** This class applies an adaptation strategy to a base VLM, passed via the `base_vlm` argument. Given an adaptation dataset `adaptation_data`, a tuning strategy specified by the `strategy` parameter, and a fixed number of training steps `num_steps`, the adapter trains the model by iteratively sampling a batch, computing the loss via `compute_loss`, and applying updates with `update_model`.

```python
class AdaptationDatasetBuilder:
    def __init__(self, raw_data_path):
        self.samples = load_ground_truth_data(raw_data_path)

    def build(self, alpha_task, beta_format, context_length,
    ↪  sampling_strategy):
        formatted = stratified_sample(
            samples=self.samples,
            task_proportions=alpha_task,
            format_proportions=beta_format
        )
        dataset = []
        for sample in formatted:
            visual_ctx = self._sample_visual_context(
                sample["frames"], context_length, sampling_strategy
            )
            dataset.append({
                "frames": visual_ctx,
                "question": sample["question"],
                "answer": sample["answer"]
            })
        return dataset

class VLMAdapter:
    def __init__(self, base_vlm):
        self.base_vlm = base_vlm

    def adapt(self, adaptation_data, strategy, num_steps):
        configure_adaptation(self.base_vlm, strategy)
        for step in range(num_steps):
```

```
                    batch = sample_from(adaptation_data)
                    loss = compute_loss(self.base_vlm, batch)
                    update_model(self.base_vlm, loss)
            return self.base_vlm
```

**Evaluation Pipeline.**

The evaluation stage can be implemented via two additional modules: `RolloutsGenerator` and `Universe`.

**`RolloutsGenerator`.** This component autoregressively samples rollout trajectories from a world model (`textttworld_model`). Given an initial observation `o_initial` and an action sequence `a_seq`, the `rollout` method generates a sequence of predicted observations by maintaining lists of past observations (`o_lt`) and actions (`a_lt`). At each timestep, it calls `predict_next_observation` to obtain the next predicted frame, appends it to the rollout sequence `o_seq`, and continues until `timestamps` is reached. This process produces a full trajectory simulating environment dynamics.

**`Universe`.** This module serves as the inference engine of our framework. It wraps an adapted VLM passed via `adapted_vlm`. Given a generated rollout and an evaluation specification, the method `evaluate_rollout` constructs a prompt using `generate_question`, parameterized by a recognition `target` and `complexity` level. It then calls `evaluate`, which queries the VLM with the resulting `rollout` and `question`, returning the model's answer.

```python
class RolloutsGenerator:
    def __init__(self):
        self.world_model = WorldModel(...)

    def predict_next_observation(self, o_lt, a_lt):
        return self.world_model(o_lt, a_lt)

    def rollout(self, o_initial, a_seq, timestamps):
        o_seq = [o_initial]
        o_lt, a_lt = [o_initial], []
        for t in range(timestamps):
            a_lt.append(a_seq[t])
            o_t = self.predict_next_observation(o_lt, a_lt)
            o_seq.append(o_t)
            o_lt.append(o_t)
        return o_seq

class Universe:
    def __init__(self, adapted_vlm):
        self.vlm = adapted_vlm

    def evaluate(self, rollout, question):
        return self.vlm(rollout, question)

    def evaluate_rollout(self, rollout, target, complexity):
        question = generate_question(rollout, target, complexity)
        return self.evaluate(rollout, question)
```

## C.2 ADAPTATION TO NEW DOMAINS

To adapt UNIVERSE to a new environment, one could collect a small set of reference trajectories that provide ground-truth observations and actions for the evaluation dimensions of interest. Question and answer templates can then be instantiated from these ground-truth signals to construct a mixed-supervision adaptation dataset.

In practice, three complementary routes exist for obtaining such reference data: (i) *Environment-side metadata*: partnering with environment developers (as in our work) provides high-fidelity ground truth through controller logs and environment state. This route yields the highest-quality labels but requires coordination with studios or simulation platforms. (ii) *Manual annotation*: when metadata is unavailable, a one-time human annotation effort can label a small set of trajectories. While this incurs upfront cost, it is less expensive than continuous human evaluation of every generated rollout. (iii) *Synthetic data generation*: for domains with publicly available gameplay videos or recordings, generative models can provide initial noisy labels. These synthetic labels can bootstrap the adaptation dataset, either directly or after human verification of a subset.

Once reference data is obtained through either route, question-answer templates are instantiated from ground-truth signals, and the resulting dataset can be used for model adaptation. Once adapted, the evaluator could provide dimension-wise assessments that can be aggregated or used as feedback in downstream world-model training.

Although our experiments use 14-frame inputs, this is a design choice rather than a fundamental constraint. The same procedure extends to longer rollouts by increasing the visual context window or applying a sliding-window scheme over successive 14-frame segments.

Overall, deploying UNIVERSE as a standalone evaluator in novel domain would require: (i) a small adaptation dataset of reference trajectories with ground-truth labels, (ii) the task specification, and (iii) the lightweight fine-tuning recipe.

## C.3 INFERENCE

At inference time, the evaluator processes rollouts by emitting natural-language responses along specified dimensions. These outputs can be used either as (i) structured feedback for world model improvement or (ii) mapped to numerical scores and aggregated into trajectory-level quality metrics.

Specifically, our method supports two complementary evaluation setups:

(i) *Ground-truth–aligned evaluation*: given a generated rollout $r$ and ground truth information $GT$ (e.g., actions conditioning the world model that we want to evaluate), we use $GT$ to instantiate $QA$ pairs with known correct answers $(A, \hat{A})$. UNIVERSE generates predictions $\hat{A}$ for each question $Q$ and we compute EM and ROUGE for $(A, \hat{A})$ to quantify alignment.

(ii) *Open-ended evaluation:* given a generated rollout $r$ and a dimension of interest (e.g., action execution, character presence), we instantiate questions from the world model's conditioning information without reference answers. UNIVERSE's responses provide structured semantic feedback, e.g., binary questions can identify which frames contain target actions or characters, enabling downstream analysis. While this setting lacks ground-truth alignment scores, the evaluator's responses can be aggregated into meaningful statistics.

# D DATASET

This section details the construction and release of the dataset used to adapt VLMs for fine-grained evaluation of world model rollouts. We curate a realistic, human-centered dataset derived from actual gameplay in a complex multi-agent environment. Designed to provide temporally grounded and semantically structured supervision, the dataset aligns with the downstream evaluation setting and supports adaptation to both action and character recognition tasks across all QA formats. We describe the data construction pipeline, QA generation process, and release format below.

## D.1 CONSTRUCTION PROCESS

The ground truth dataset for adapting the evaluator (see Section 3) was developed in collaboration with *Ninja Theory* using human gameplay recordings from *Bleeding Edge*, a 4v4 multiplayer combat game. Data use was governed by a formal agreement with the studio, and collection adhered to the game's End User License Agreement (EULA). All protocols were approved by our Institutional Review Board (IRB), and personally identifiable information (PII) was removed prior to analysis.

Each gameplay session is represented as a tuple $s = (v, c, m)$, where $v$ is a high-resolution MP4 video (60 FPS), $c$ is the synchronized controller action log, and $m$ contains structured metadata (e.g., player roles, agent identities, action categories, and map context). The full set of gameplay sessions is denoted by $\mathcal{S} = \{(v_i, c_i, m_i)\}_{i=1}^{|\mathcal{S}|}$.

The dataset construction pipeline proceeds in three stages:

(i) *Preprocessing.* We begin by filtering out corrupted applying or inactive sessions and synchronizes the video, controller logs, and metadata streams using internal game times-tamps: $\mathcal{S}_{\text{valid}} = \text{Preprocessing}(\mathcal{S})$. Each valid session is segmented into non-overlapping clips of fixed length $L = 14$ frames, each paired with controller input and shared metadata; formally, for a session $s = (v, c, m) \in \mathcal{S}_{\text{valid}}$, the segmentation pro-duces $\text{Segment}(v, c, m, L) = \{(f^{(1:L)}, c^{(1:L)}, m)\}$, where $f^{(1:L)}$ denotes the sequence of frames, $c^{(1:L)}$ the aligned controller inputs, and $m$ the associated metadata. The complete set of extracted clips across all valid sessions is defined as $\mathcal{V} = \bigcup_{s \in \mathcal{S}_{\text{valid}}} \text{Segment}(s, L)$, where each element $v \in \mathcal{V}$ is a triplet $(f^{(1:L)}, c^{(1:L)}, m)$ consisting of video frames, corresponding controller inputs, and metadata.

(ii) *Description Generation.* Next, for each sequence of frames $f^{(1:L)} \in \mathcal{V}$, we use the associated control log $c^{(1:L)}$ to extract action information and the metadata $m$ to obtain character-related attributes. These are combined to generate a structured natural language description via $d = \text{Describe}(c^{(1:L)}, m)$, where $\text{Describe}$ is a rule-based procedure that transforms the logged actions and metadata into textual descriptions used for constructing the QA su-pervision. This yields a set of paired video–text examples: $\mathcal{Z} = \{(f^{(1:L)}, d) \mid f^{(1:L)} \in \mathcal{V}\}$.

(iii) *Question-Answer Pair Construction.* Finally, we generate six QA pairs per clip, spanning two predefined tasks (AR and CR), each instantiated in three question formats: binary, multiple-choice, and open-ended. To enable this, we define task-specific answer spaces us-ing $\text{GetAnswerSpace}(\mathcal{Z})$, which returns $\mathcal{Y}_{\text{AR}}$ for action categories and $\mathcal{Y}_{\text{CR}}$ for character identities, based on all video–text pairs in $\mathcal{Z}$. For each clip, we extract the task-specific ground-truth answer from the corresponding description as $y = \text{ExtractLabel}(d, t)$, where $t \in \{\text{AR}, \text{CR}\}$. Each QA format is constructed as follows: (i) *Binary*: Two binary question-answer pairs are generated per instance using $\text{FormatBinaryPrompt}$. The positive question $Q^{\text{pos}}$ is constructed using the correct label $y \in \mathcal{Y}^{(t)}$ and paired with the positive answer $A^{\text{pos}}$. The negative question $Q^{\text{neg}}$ is constructed using an incorrect label $\tilde{y} \sim \text{SampleDistractor}(\mathcal{Y}^{(t)} \setminus \{y\})$ and paired with the negative answer $A^{\text{neg}}$. (ii) *Mul-tiple-Choice*: A question $Q$ is generated using the full set of candidate options, formatted via $\text{FormatOptions}(\mathcal{Y}_t)$. The question is constructed with $\text{FormatMCPrompt}(t, O)$ and paired with the correct answer $y \in \mathcal{Y}_t$. (iii) *Open-Ended*: A free-form question $Q$ is generated using $\text{FormatOEPrompt}(t)$, prompting the model to produce the correct label $y \in \mathcal{Y}_t$ without access to predefined answer choices.

The final dataset is represented as $\mathcal{D} = \{(f_i^{(1:L)}, QA_i)\}_{i=1}^{|\mathcal{D}|}$, where each $f^{(1:L)}$ is a video clip and $QA = \{(Q_j, A_j)\}_{j=1}^{6}$ is the associated set of question–answer pairs, covering all combinations of three question formats (binary, multiple-choice, open-ended) and two tasks (Action Recognition and Character Recognition). A detailed data pipeline is provided in Algorithm 1.

---

**Algorithm 1** Dataset Construction Process

---

**Procedure** `DatasetCreation`$(\mathcal{S}, L)$**:**

$\quad \mathcal{S}_{\text{valid}} \leftarrow$ `Preprocessing`$(\mathcal{S})$

$\quad \mathcal{V} \leftarrow \emptyset$

$\quad$ **for** $(v, c, m) \in \mathcal{S}_{valid}$ **do**

$\quad\quad \mathcal{V}_s \leftarrow$ `Segment`$(v, c, m, L)$

$\quad\quad \mathcal{V} \leftarrow \mathcal{V} \cup \mathcal{V}_s$

$\quad \mathcal{Z} \leftarrow \emptyset$

$\quad$ **for** $(f^{(1:L)}, c^{(1:L)}, m) \in \mathcal{V}$ **do**

$\quad\quad d \leftarrow$ `Describe`$(m, c^{(1:L)})$

$\quad\quad \mathcal{Z} \leftarrow \mathcal{Z} \cup \{(f^{(1:L)}, d)\}$

$\quad \mathcal{D} \leftarrow \emptyset$

$\quad \mathcal{Y}_{\text{AR}}, \mathcal{Y}_{\text{CR}} \leftarrow$ `GetAnswerSpace`$(\mathcal{Z})$

$\quad$ **for** $(f^{(1:L)}, d) \in \mathcal{V}$ **do**

$\quad\quad \mathcal{QA} \leftarrow$ `GenerateQAPairs`$(d, \mathcal{Y}_{\text{AR}}, \mathcal{Y}_{\text{CR}})$

$\quad\quad$ **for** $(Q, A) \in \mathcal{QA}$ **do**

$\quad\quad\quad \mathcal{D} \leftarrow \mathcal{D} \cup \{(f^{(1:L)}, Q, A)\}$

$\quad$ **return** $\mathcal{D}$

**Procedure** `GenerateQAPairs`$(d, \mathcal{Y}_{CR}, \mathcal{Y}_{CR})$**:**

$\quad \mathcal{QA} \leftarrow \emptyset$

$\quad$ **for** $t \in \{AR, CR\}$ **do**

$\quad\quad y \leftarrow$ `ExtractLabel`$(d, t)$

$\quad\quad QA_{\text{bin}}^{pos}, QA_{\text{bin}}^{neg} \leftarrow$ `CreateBinaryQA`$(t, y)$

$\quad\quad \mathcal{QA} \leftarrow \mathcal{QA} \cup \{QA_{\text{bin}}^{\text{pos}}, QA_{\text{bin}}^{\text{neg}}\}$

$\quad\quad QA_{\text{mc}} \leftarrow$ `CreateMCQA`$(t, y, \mathcal{Y}_t)$

$\quad\quad \mathcal{QA} \leftarrow \mathcal{QA} \cup QA_{\text{mc}}$

$\quad\quad QA_{\text{oe}} \leftarrow$ `CreateOpenEndedQA`$(t, y)$

$\quad\quad \mathcal{QA} \leftarrow \mathcal{QA} \cup QA_{\text{oe}}$

$\quad$ **return** $\mathcal{Q}$

**Procedure** `CreateBinaryQA`$(t, y)$**:**

$\quad \tilde{y} \leftarrow$ `SampleDistractor`$(\mathcal{Y}_t \setminus \{y\})$

$\quad Q^{pos} \leftarrow$ `FormatBinaryPrompt`$(t, y)$

$\quad Q^{neg} \leftarrow$ `FormatBinaryPrompt`$(t, \tilde{y})$

$\quad$ **return** $\{(Q^{pos}, A^{pos}), (Q_{\tilde{y}}, A^{neg})\}$

**Procedure** `CreateMCQA`$(t, y, \mathcal{Y}_t)$**:**

$\quad O \leftarrow$ `FormatOptions`$(\mathcal{Y}_t)$

$\quad Q \leftarrow$ `FormatMCPrompt`$(t, O)$

$\quad$ **return** $Q, y$

**Procedure** `CreateOpenEndedQA`$(t, y)$**:**

$\quad Q \leftarrow$ `FormatOEPrompt`$(t)$

$\quad$ **return** $Q, y$

---

## D.2 RELEASE DETAILS

To support reproducibility and further research, we aim to release a subset of our evaluation data. This includes sampled human gameplay segments, aligned action vectors and environment states, natural language descriptions, and QA annotations spanning binary, multiple-choice, and open-ended formats. The dataset is included in the supplementary ZIP file and will be publicly released following the publication of the paper.

**File Layout.** The data is organized as follows:

- `human-gameplay-segments/`: directory of `.npz` files, each containing image frames along with frame-aligned actions and states;

- `annotations.jsonl`: line-delimited JSON file containing natural language descriptions, QA prompts, and ground truth answers.

**Structure.** Each dataset instance corresponds to a short human gameplay segment stored as a NumPy archive (`.npz`), containing:

(i) `images` $\in \mathbb{R}^{14 \times 3 \times 180 \times 300}$: a sequence of 14 RGB frames in channel-first (CHW) format;

(ii) `actions` $\in \mathbb{R}^{14 \times 16}$: frame-aligned control vectors;

(iii) `states` $\in \mathbb{R}^{14 \times 56}$: frame-aligned environment states.

**Annotation Format.** Annotations are provided in `annotations.jsonl`, a line-delimited JSON file where each entry corresponds to a single gameplay segment. Each entry includes structured prompts and ground truth answers spanning all tasks and formats.

Specifically, each annotation entry includes:

- `filename`: Unique identifier of the associated `.npz` file containing visual observations (frames), action vectors, and states.
- `description`: Natural language summary of the video segment.
- `ar_binary_pos_q`, `ar_binary_pos_a`: Affirmative binary question and corresponding answer, evaluating recognition of the correct action.
- `ar_binary_neg_q`, `ar_binary_neg_a`: Negative binary question and corresponding answer, targeting rejection of an incorrect action.
- `ar_mc`: Multiple-choice question prompting the model to select the correct action from a list of candidate classes.
- `ar_oe`: Open-ended question prompting free-form generation of the observed action.
- `ar_answer`: Ground truth action label corresponding to both `ar_mc` and `ar_oe`.
- `cr_binary_pos_q`, `cr_binary_pos_a`: Affirmative binary question and corresponding answer for identifying the correct character.
- `cr_binary_neg_q`, `cr_binary_neg_a`: Negative binary question and corresponding answer targeting an incorrect character identity.
- `cr_mc`: Multiple-choice question prompting identification of the correct character from a candidate set.
- `cr_oe`: Open-ended question prompting free-form naming of the character.
- `cr_answer`: Ground truth character label shared across both `cr_mc` and `cr_oe`.

# E  EXPERIMENTAL DETAILS

In this section, we provide a detailed description of the dataset preparation process, model architecture, prompt templates, training procedure. Additionally, we provide an overview of all results presented in the main paper in numerical table form, an report additional experimental results leveraging alternate fine-tuning solutions.

## E.1  MODEL

This section provides extended details on the architecture, pretraining configuration, and input formatting of the vision-language models used in our experiments. Our primary backbone is PaliGemma (Beyer et al., 2024; Steiner et al., 2024).

### E.1.1  OVERVIEW

PaliGemma is a VLM that processes both images and text as input and autoregressively generates natural language output. It follows the training paradigm of PaLI-3 (Chen et al., 2023), combining a ViT-based vision encoder (Dosovitskiy et al., 2021) with a decoder-only Transformer language model.

Table 1: Detailed architecture of the PaliGemma model, comprising a SigLIP-So400m vision tower, a multimodal projection head, and a Gemma-based language decoder. All transformer layers follow standard design and include residual connections around attention and MLP blocks.

| Component | Configuration |
|---|---|
| *Vision Tower: SigLIP-So400m* | |
| Patch Embedding | Conv2d(in=3, out=1152, kernel=14, stride=14) |
| Position Embedding | Embedding(num_embeddings=256, emb_dim=1152) |
| Encoder | 27 × Transformer Encoder Layers |
| Self-Attention | — |
| Query / Key / Value projection | Linear(1152 → 1152, bias=True) |
| Layer Normalization | LayerNorm((1152,), eps=1e-6) |
| MLP Block | — |
| Activation Function | GELU-Tanh |
| Feedforward layer (up) | Linear(1152 → 4304, bias=True) |
| Feedforward layer (down) | Linear(4304 → 1152, bias=True) |
| Layer Normalization | LayerNorm((1152,), eps=1e-6) |
| Post-Encoder Layer Norm | LayerNorm((1152,), eps=1e-6) |
| *Multimodal Projection Head* | |
| Linear Projection | Linear(1152 → 2304, bias=True) |
| *Language Model: Gemma* | |
| Token Embedding | Embedding(vocab=257216, dim=2304) |
| Decoder Stack | 26 × Transformer Decoder Layers |
| Self-Attention | — |
| Query projection | Linear(2304 → 2048, bias=False) |
| Key projection | Linear(2304 → 1024, bias=False) |
| Value projection | Linear(2304 → 1024, bias=False) |
| Output projection | Linear(2048 → 2304, bias=False) |
| MLP Block | — |
| Gating projection | Linear(2304 → 9216, bias=True) |
| Down projection | Linear(2304 → 9216, bias=True) |
| Up projection | Linear(9216 → 2304, bias=True) |
| Activation Function | GELU-Tanh |
| Normalization Layers | — |
| Input Norm | RMSNorm(2304, eps=1e-6) |
| Post-Attn Norm | RMSNorm(2304, eps=1e-6) |
| Pre-FFN Norm | RMSNorm(2304, eps=1e-6) |
| Post-FFN Norm | RMSNorm(2304, eps=1e-6) |
| Rotary Embeddings | GemmaRotaryEmbedding |
| LM Head | Linear(2304 → 257216, bias=False) |

The model is publicly available (Wolf et al., 2019). The architecture is fully modular, comprising three parameterized components: (i) *Vision encoder* ($\mathcal{M}_V$): based on SigLIP (Zhai et al., 2023), specifically the "shape optimized" So400m (Alabdulmohsin et al., 2023). (ii) *Multimodal projection head* ($\mathcal{M}_P$): a single linear layer for projecting visual features into the language decoder's embedding space. (iii) *Language decoder* ($\mathcal{M}_L$): a Transformer-based autoregressive model from the Gemma family (Mesnard et al., 2024; Rivière et al., 2024). Below, we discuss the architecture in more details, the general layer-level overview is also provided in Table 1.

**Vision Encoder: SigLIP-So400m.** The visual backbone $\mathcal{M}_V$ is a ViT-style encoder pretrained using a Sigmoid contrastive loss (SigLIP). It processes input images by dividing them into non-overlapping $14 \times 14$ patches. Each patch is linearly projected into a 1152-dimensional embedding via a convolutional stem. To encode spatial structure, learned positional embeddings are added before

Table 2: Component-wise parameter overview of the PaliGemma model.

| Component | Model / Variant | Details | # Params |
|-----------|-----------------|---------|----------|
| Vision Encoder | SigLIP-So400m | Input resolutions: $224\text{px}^2$, $448\text{px}^2$, $896\text{px}^2$ | 400M |
| Multimodal Projection | — | Connects vision and language components | 2.66M |
| Language Model | PG 1
PG 2
PG 3 | Gemma 1 2B, pre-trained on 6T tokens
Gemma 2 2B, pre-trained on 2T tokens
Gemma 2 9B, pre-trained on 8T tokens | 3B
3B
9.7B |

the representation is passed through a stack of 27 SigLIP encoder layers. Each encoder layer contains multi-head self-attention with projection layers for queries, keys, and values, followed by an MLP block with GELU-Tanh activations. All transformer blocks use LayerNorm and residual connections. The vision tower supports multiple input resolutions (224, 448, 896), though our experiments fix resolution at $224\text{px}^2$ for consistency and efficiency.

**Multimodal Projection Head.** The projection head $\mathcal{M}_P$ is a lightweight linear mapping from the vision encoder's output dimension (1152) to the language decoder's input dimension (2304). It contains approximately 2.66M parameters and is initialized with zero-mean weights. This head enables alignment between visual and linguistic modalities and is important for bridging the representation gap between the vision and language components.

**Language Decoder: Gemma.** The language module $\mathcal{M}_L$ is a decoder-only Transformer with 26 layers and 2304-dimensional hidden states. Token embeddings are learned over a vocabulary of 257,216 tokens, encoded using the SentencePiece tokenizer (Kudo & Richardson, 2018). Each Transformer block contains a self-attention mechanism with separate linear projections for queries, keys, and values. The MLP block follows a gated architecture, where the input is processed through parallel down projection and gating projection layers, modulated by a GELU-Tanh activation (Hendrycks & Gimpel, 2016), combined via elementwise multiplication, and then passed through an up projection to return to the model's hidden dimension. RMSNorm is applied before and after both attention and MLP sublayers to stabilize training. Rotary positional embeddings are added to enable relative position encoding. Output tokens are produced via a tied language modeling head that projects back to the vocabulary space.

### E.1.2 CONFIGURATIONS

Table 2 summarizes the architecture components and parameter counts of the PaliGemma configurations available for experimentation. While we focus on the PaliGemma 2 3b variant in our study, we include all publicly released configurations for completeness and to clarify how our selected model compares to other available options. All three variants share the same vision encoder and multimodal integration strategy, differing only in the language decoder. The first configuration, PaliGemma 1 3b, pairs the visual encoder with Gemma 1 (2B), pretrained on 6 trillion tokens, resulting in a total model size of approximately 3 billion parameters. The second configuration, PaliGemma 2 3b, replaces the decoder with Gemma 2 (2B), pretrained on 2 trillion tokens, and maintains a comparable total parameter count. The third and largest variant, PaliGemma 2 10b, uses Gemma 2 (9B) as the decoder, pretrained on 8 trillion tokens, yielding a total model size of approximately 9.7 billion parameters.

### E.1.3 PROMPT FORMAT

To generate textual responses, we adopt a unified prompt format for the decoder. Each input sequence consists of image tokens $S_{\mathcal{I}}$, a textual prefix $S_{\mathcal{T}}^{\text{PREF}}$ containing the question, and a suffix $S_{\mathcal{T}}^{\text{SUFF}}$ containing the expected answer. The model autoregressively generates the answer tokens, and training loss is applied only to the suffix.

Let $n$ denote the number of input frames and $p$ the number of visual tokens (patch embeddings) per frame. In our setting, each frame is encoded as $p = 256$ visual tokens. The overall input schema is as follows:

$$S = \underbrace{\texttt{<image>}_1^{(1)}, \ldots, \texttt{<image>}_p^{(1)}, \ldots, \texttt{<image>}_1^{(n)}, \ldots, \texttt{<image>}_p^{(n)}}_{S_{\mathcal{I}}: \text{ Visual tokens from } n \text{ frames, each represented as } p \text{ patches}}$$

$$\underbrace{\texttt{<BOS>, answer en, <QUESTION>, <SEP>}}_{S_{\mathcal{T}}^{\text{PREF}}: \text{ Prefix (cue + question)}}$$

$$\underbrace{\texttt{<ANSWER>, <EOS>, <PAD>,\ldots,<PAD>}}_{S_{\mathcal{T}}^{\text{SUFF}}: \text{ Suffix (answer)}}$$

Here, $S_{\mathcal{I}}$ contains visual tokens produced by the vision encoder $\mathcal{M}_V$, and projected into $\mathcal{M}_L$ space using $\mathcal{M}_P$. The prefix $S_{\mathcal{T}}^{\text{PREF}}$ starts with a special `<BOS>` token and includes a task-language cue (e.g., "`answer en`"), the question, and a separator `<SEP>`. The suffix $S_{\mathcal{T}}^{\text{SUFF}}$ contains the target answer, terminated with `<EOS>` and padded with `<PAD>` tokens for batching.

### E.1.4 PRETRAINING DATA AND FILTERING

PaliGemma is pretrained on a mixture of large-scale vision-language datasets, including WebLI (Chen et al., 2022b), CC3M-35L (Sharma et al., 2018), VQ$^2$A-CC3M-35L (Changpinyo et al., 2022), OpenImages (Piergiovanni et al., 2022), and WIT (Srinivasan et al., 2021). Data quality and safety are maintained through pornographic content filtering, text safety and toxicity filtering, and privacy-preserving measures.

### E.2 EVALUATION METRICS

In this section, we provide additional details on metrics used for quantitative evaluation. We employ two complementary metrics: *Exact Match (EM)* and *ROUGE-F$_1$ (ROUGE)*, which together capture both syntactic precision and semantic alignment.

**Exact Match Accuracy ($EM$)** measures whether the generated answer is identical to the expected answer, providing a high-precision signal for correctness. Formally, it is defined as:

$$EM = \mathbb{1}(\hat{A} = A) \tag{2}$$

where $\hat{A}$ is the model's prediction and $A$ is the corresponding ground-truth answer. This metric is especially informative for binary and multiple-choice formats where the output space is well-defined.

**ROUGE F$_1$ (ROUGE)** captures token-level semantic overlap between generated and reference responses by computing the harmonic mean of precision and recall. This allows us to account for partially correct or paraphrased answers. For binary questions, we compute the metric on the bigram level, while for multiple-choice and open-ended formats, we use trigram-level evaluation.

Formally, let $G$ and $GT$ denote the sets of $n$-grams in the generated and ground truth answers, respectively. Precision and recall are defined as:

$$P = \frac{|G \cap GT|}{|G|}, \quad R = \frac{|G \cap GT|}{|GT|} \tag{3}$$

where $|G \cap GT|$ counts overlapping $n$-grams. The ROUGE score is then computed as:

$$\text{ROUGE} = 2 \times \frac{P \times R}{P + R} \tag{4}$$

Together, these metrics provide a robust view of model performance: EM reflects exact correctness, while ROUGE provides a softer measure of semantic fidelity, particularly useful for evaluating open-ended generations.

### E.3 RESULTS

**Hyperparameters.** Table 3 summarizes the core training hyperparameters used across all adaptation experiments. We train all models on 8 NVIDIA A100 GPUs with a batch size of 1 per device and

Table 3: Summary of hyperparameters used in our experiments.

| Hyperparameter | Value |
|---|---|
| Input resolution | $224 \times 224$ |
| Image frames per input | 1–8 |
| Number of epochs | 1–10 |
| Batch size (per device) | 1 |
| Gradient accumulation steps | 4 |
| Optimizer | AdamW Loshchilov & Hutter (2019) |
| Learning rate | $5 \times 10^{-5}$, cosine annealing |
| Learning rate warmup | 10% |
| Weight decay | $1 \times 10^{-6}$ |
| Gradient clipping | Global norm, threshold 1.0 |
| VLM backbone | PaliGemma 2 (3B) Beyer et al. (2024) |

accumulate gradients over 4 steps, yielding an effective batch size of 32. Each epoch corresponds to a full pass over the adaptation dataset, and no early stopping is applied. Models were trained for 1–10 epochs depending on task and setting. Optimization is performed using AdamW (Loshchilov & Hutter, 2019) with parameters $\beta_1 = 0.9$, $\beta_2 = 0.999$, a base learning rate of $5 \times 10^{-5}$, and weight decay of $1 \times 10^{-6}$. We use cosine learning rate annealing (Loshchilov & Hutter, 2022) with a linear warmup over the first 10% of training steps. To stabilize training, we apply gradient clipping with a global norm threshold of 1.0. All models use PaliGemma 2 (3B) (Beyer et al., 2024) as the vision-language backbone unless otherwise noted. We vary the number of input frames between 1 and 8 depending on task, and all images are resized to a fixed resolution of $224 \times 224$. Training is conducted in `bfloat16` precision using data parallelism. Model selection is based on final validation accuracy.

**Tabular Results Summary.** The following tables summarize primary experimental findings across our study. Each entry corresponds to a core evaluation or analysis in the paper, organized by experimental section and aligned with the corresponding table description.

- *Zero-Shot Evaluation* (Section 4): Table 4 reports ROUGE-$F_1$ zero-shot performance of pretrained PaliGemma and VideoLLaMA3 models on Action and Character Recognition tasks. Models are evaluated in a zero-shot setting with 1 or 8 input frames, across binary, multiple-choice, and open-ended formats.

- *Fine-Tuned Baselines* (Section 4): Table 5 reports ROUGE-$F_1$ and Exact Match performance of PaliGemma 2 variants fine-tuned using full, partial, and parameter-efficient strategies. All models are trained on a single frame for one epoch, and evaluated across binary, multiple-choice, and open-ended formats.

- *Analysis: Supervision and Temporal Context* (Section 5): Table 6 examines early-stage learning dynamics on Character Recognition (CR), with evaluation at sub-epoch intervals. Table 7 reports AR performance as a function of training budget, scaling the number of epochs with a single input frame. Table 8 extends this analysis to jointly vary training epochs and the number of input frames, disentangling the effects of temporal context and supervision on AR.

- *Analysis: Temporal Sampling Strategies* (Section 5): Table 9 compares first-$n$ and uniform-$n$ frame sampling strategies for Action Recognition, evaluating model performance across varying temporal context lengths ($n \in \{1, \ldots, 8\}$).

- *Analysis: Optimizing Data Mix for Unified Multi-Task Evaluation* (Section 5): This analysis spans three tables. Table 10 explores task-level trade-offs when jointly training on Action and Character Recognition by varying $\alpha_{\mathrm{AR}}$ vs. $\alpha_{\mathrm{CR}}$, with format distribution held uniform. Table 11 fixes $\alpha_{\mathrm{AR}} = 0.8$ and searches over format-level ratios ($\beta$), revealing the impact of increased open-ended (OE) supervision. Table 12 further investigates this high-OE regime, balancing the remaining budget between binary and multiple-choice for optimal performance.

Table 4: Zero-shot ROUGE-$F_1$-based evaluation of PaliGemma (PG) and VideoLLaMA3 (VL3) models on Action and Character Recognition tasks using 1 and 8 input frames. "MC" denotes multiple-choice and "OE" open-ended formats.

| Fr | Model | Action Recognition | | | Character Recognition | | |
|----|-------|--------|------|------|--------|------|------|
| | | Binary | MC | OE | Binary | MC | OE |
| 1 | PG 1 3B | $50.43 \pm 0.13$ | $8.12 \pm 0.02$ | $10.83 \pm 0.01$ | $50.73 \pm 0.38$ | $0.46 \pm 0.06$ | $0.00 \pm 0.00$ |
| | PG 2 3B | $44.69 \pm 0.03$ | $9.30 \pm 0.17$ | $12.64 \pm 0.01$ | $48.58 \pm 0.07$ | $0.28 \pm 0.06$ | $0.01 \pm 0.00$ |
| | PG 2 10B | $50.04 \pm 0.03$ | $26.98 \pm 0.00$ | $12.35 \pm 0.21$ | $50.08 \pm 0.07$ | $8.33 \pm 0.50$ | $0.00 \pm 0.00$ |
| | VL3-2B | $3.24 \pm 0.00$ | $18.52 \pm 0.06$ | $6.27 \pm 0.05$ | $8.76 \pm 0.04$ | $3.44 \pm 0.08$ | $0.50 \pm 0.01$ |
| | VL3-7B | $45.02 \pm 0.28$ | $15.53 \pm 0.05$ | $6.54 \pm 0.04$ | $39.09 \pm 0.73$ | $6.21 \pm 0.05$ | $0.51 \pm 0.02$ |
| 8 | PG 1 3B | $51.67 \pm 0.02$ | $10.68 \pm 0.00$ | $10.32 \pm 0.00$ | $51.39 \pm 0.07$ | $0.25 \pm 0.00$ | $0.00 \pm 0.00$ |
| | PG 2 3B | $47.61 \pm 0.19$ | $6.73 \pm 0.04$ | $14.52 \pm 0.00$ | $48.37 \pm 0.18$ | $0.03 \pm 0.00$ | $0.01 \pm 0.00$ |
| | PG 2 10B | $50.02 \pm 0.06$ | $26.93 \pm 0.01$ | $12.12 \pm 0.00$ | $50.09 \pm 0.06$ | $0.22 \pm 0.00$ | $0.00 \pm 0.00$ |
| | VL3-2B | $13.92 \pm 0.13$ | $3.47 \pm 0.02$ | $0.32 \pm 0.01$ | $13.92 \pm 0.13$ | $3.46 \pm 0.04$ | $0.32 \pm 0.01$ |
| | VL3-7B | $15.05 \pm 0.21$ | $16.67 \pm 0.35$ | $6.35 \pm 0.06$ | $12.76 \pm 0.52$ | $5.88 \pm 0.01$ | $0.54 \pm 0.01$ |

Table 5: Performance of fine-tuned PaliGemma 2 variants on Action and Character Recognition tasks. We compare full, partial, and parameter-efficient tuning strategies. "MC" denotes multiple-choice and "OE" open-ended formats.

| Model | Binary | | Multiple-choice | | Open-ended | |
|-------|--------|-------|-----------------|-------|------------|-------|
| | EM | ROUGE | EM | ROUGE | EM | ROUGE |
| | | | Action Recognition | | | |
| $\mathcal{F}_L$ | $50.00 \pm 0.00$ | $50.00 \pm 0.00$ | $13.13 \pm 0.00$ | $27.57 \pm 0.00$ | $13.13 \pm 0.00$ | $27.57 \pm 0.00$ |
| $\mathcal{F}_P$ | $83.97 \pm 0.02$ | $83.97 \pm 0.02$ | $61.43 \pm 0.58$ | $68.05 \pm 0.70$ | $61.68 \pm 0.35$ | $68.46 \pm 0.19$ |
| $\mathcal{F}_V$ | $83.70 \pm 0.97$ | $83.70 \pm 0.97$ | $63.40 \pm 0.45$ | $69.87 \pm 0.44$ | $66.03 \pm 0.10$ | $71.92 \pm 0.08$ |
| $\mathcal{F}_{P+L}$ | $74.47 \pm 1.64$ | $74.47 \pm 1.64$ | $13.13 \pm 0.00$ | $27.57 \pm 0.00$ | $55.74 \pm 0.70$ | $64.83 \pm 0.29$ |
| $\mathcal{F}_{V+L}$ | $75.80 \pm 0.16$ | $75.80 \pm 0.16$ | $13.13 \pm 0.00$ | $27.57 \pm 0.00$ | $13.13 \pm 0.00$ | $27.57 \pm 0.00$ |
| $\mathcal{F}_{V+P}$ | $73.46 \pm 0.85$ | $73.46 \pm 0.85$ | $61.21 \pm 0.23$ | $67.57 \pm 0.21$ | $64.70 \pm 0.02$ | $70.93 \pm 0.01$ |
| $\mathcal{F}_{all}$ | $74.35 \pm 1.37$ | $74.35 \pm 1.37$ | $13.13 \pm 0.00$ | $27.57 \pm 0.00$ | $13.13 \pm 0.00$ | $27.57 \pm 0.00$ |
| $\mathcal{F}_{LoRA}$ | $44.66 \pm 0.21$ | $44.66 \pm 0.21$ | $0.02 \pm 0.01$ | $9.21 \pm 0.01$ | $0.00 \pm 0.00$ | $12.49 \pm 0.00$ |
| | | | Character Recognition | | | |
| $\mathcal{F}_L$ | $50.00 \pm 0.00$ | $50.00 \pm 0.00$ | $98.92 \pm 0.00$ | $98.92 \pm 0.00$ | $98.98 \pm 0.00$ | $98.99 \pm 0.01$ |
| $\mathcal{F}_P$ | $99.09 \pm 0.11$ | $99.09 \pm 0.11$ | $99.22 \pm 0.33$ | $99.22 \pm 0.33$ | $99.15 \pm 0.07$ | $99.15 \pm 0.07$ |
| $\mathcal{F}_V$ | $99.31 \pm 0.01$ | $99.31 \pm 0.01$ | $99.14 \pm 0.42$ | $99.14 \pm 0.42$ | $99.61 \pm 0.12$ | $99.61 \pm 0.12$ |
| $\mathcal{F}_{P+L}$ | $50.00 \pm 0.00$ | $50.00 \pm 0.00$ | $98.28 \pm 0.00$ | $98.30 \pm 0.02$ | $98.39 \pm 0.00$ | $98.39 \pm 0.00$ |
| $\mathcal{F}_{V+L}$ | $50.00 \pm 0.00$ | $50.00 \pm 0.00$ | $96.88 \pm 0.00$ | $96.88 \pm 0.00$ | $98.45 \pm 0.00$ | $98.45 \pm 0.00$ |
| $\mathcal{F}_{V+P}$ | $60.32 \pm 0.02$ | $60.32 \pm 0.02$ | $99.22 \pm 0.00$ | $99.22 \pm 0.00$ | $99.79 \pm 0.00$ | $99.79 \pm 0.00$ |
| $\mathcal{F}_{all}$ | $50.00 \pm 0.00$ | $50.00 \pm 0.00$ | $97.67 \pm 0.06$ | $97.67 \pm 0.06$ | $96.55 \pm 0.01$ | $96.55 \pm 0.01$ |
| $\mathcal{F}_{LoRA}$ | $48.76 \pm 0.00$ | $48.76 \pm 0.00$ | $0.00 \pm 0.00$ | $0.32 \pm 0.00$ | $0.00 \pm 0.00$ | $0.01 \pm 0.01$ |

# F  HUMAN ANNOTATION STUDY

This section provides full details of our human annotation study, including rollout generation, annotation procedures, inter-annotator agreement, and evaluation metrics. The goal is to validate the adapted VLM's fine-grained predictions on generated video rollouts.

## F.1  STUDY DESIGN

**Task Overview.** Human annotators were presented with short video clips generated by a world model, each paired with a natural language question and an answer generated by the VLM. They were asked to judge whether the model's answer accurately described what was shown in the video. Each QA pair was rated using one of four categories: *Correct* (score = 1), *Partially Correct* (0.5), *Incorrect* (0), or *Unclear / Cannot Tell* (excluded from accuracy computation).

Table 6: *Supervision and Temporal Context:* Training budget analysis for Character Recognition. Models are fine-tuned for sub-epoch durations and evaluated across binary, multiple-choice (MC), and open-ended (OE) formats.

| Ep | Binary | | Multiple-choice | | Open-ended | |
|---|---|---|---|---|---|---|
| | EM | ROUGE | EM | ROUGE | EM | ROUGE |
| 0.005 | 50.84 ± 1.70 | 50.84 ± 1.70 | 14.27 ± 0.00 | 14.27 ± 0.00 | 13.16 ± 0.00 | 13.27 ± 0.00 |
| 0.01 | 54.92 ± 0.84 | 54.92 ± 0.84 | 17.85 ± 0.00 | 17.85 ± 0.00 | 16.78 ± 0.14 | 16.88 ± 0.01 |
| 0.02 | 57.91 ± 3.19 | 57.91 ± 3.19 | 28.64 ± 0.00 | 28.64 ± 0.00 | 26.29 ± 2.96 | 28.38 ± 0.01 |
| 0.03 | 57.45 ± 0.58 | 57.45 ± 0.58 | 64.19 ± 0.00 | 64.19 ± 0.00 | 40.76 ± 0.01 | 40.98 ± 0.00 |
| 0.06 | 65.88 ± 3.71 | 65.88 ± 3.71 | 93.96 ± 0.00 | 93.96 ± 0.00 | 88.89 ± 0.00 | 88.95 ± 0.01 |
| 0.10 | 87.47 ± 4.11 | 87.47 ± 4.11 | 96.54 ± 0.00 | 96.54 ± 0.00 | 97.01 ± 0.38 | 97.02 ± 0.40 |
| 0.125 | 91.63 ± 7.71 | 91.63 ± 7.71 | 97.08 ± 0.00 | 97.08 ± 0.00 | 97.28 ± 0.00 | 97.30 ± 0.00 |
| 0.20 | 97.96 ± 0.45 | 97.96 ± 0.45 | 97.89 ± 0.28 | 97.89 ± 0.28 | 98.12 ± 0.00 | 98.14 ± 0.02 |
| 0.25 | 97.75 ± 0.74 | 97.75 ± 0.74 | 98.08 ± 0.00 | 98.08 ± 0.00 | 98.12 ± 0.00 | 98.15 ± 0.00 |
| 0.33 | 98.42 ± 0.20 | 98.42 ± 0.20 | 98.19 ± 0.00 | 98.19 ± 0.00 | 98.30 ± 0.00 | 98.35 ± 0.00 |
| 0.50 | 98.74 ± 0.06 | 98.74 ± 0.06 | 98.45 ± 0.00 | 98.45 ± 0.00 | 98.51 ± 0.00 | 98.54 ± 0.00 |
| 0.67 | 99.11 ± 0.10 | 99.11 ± 0.10 | 98.99 ± 0.08 | 98.99 ± 0.08 | 99.09 ± 0.00 | 99.09 ± 0.00 |
| 0.75 | 99.03 ± 0.04 | 99.03 ± 0.04 | 99.15 ± 0.00 | 99.15 ± 0.00 | 99.21 ± 0.00 | 99.22 ± 0.01 |
| 1 | 99.09 ± 0.11 | 99.09 ± 0.11 | 99.22 ± 0.33 | 99.22 ± 0.33 | 99.15 ± 0.07 | 99.15 ± 0.07 |

Table 7: *Supervision and Temporal Context:* Training budget analysis for Action Recognition, with models fine-tuned for up to 10 epochs. Evaluated using across binary, multiple-choice (MC), and open-ended (OE) formats.

| Ep | Binary | | Multiple-Choice | | Open-Ended | |
|---|---|---|---|---|---|---|
| | EM | ROUGE | EM | ROUGE | EM | ROUGE |
| 1 | 83.97 ± 0.02 | 83.97 ± 0.02 | 61.43 ± 0.58 | 68.05 ± 0.70 | 61.68 ± 0.35 | 68.46 ± 0.19 |
| 2 | 84.92 ± 0.23 | 84.92 ± 0.23 | 64.90 ± 0.15 | 71.17 ± 0.01 | 64.05 ± 0.01 | 70.36 ± 0.01 |
| 4 | 85.37 ± 0.30 | 85.37 ± 0.30 | 64.58 ± 0.69 | 70.89 ± 0.55 | 64.79 ± 0.31 | 70.88 ± 0.27 |
| 8 | 85.18 ± 0.20 | 85.18 ± 0.20 | 63.53 ± 0.35 | 69.95 ± 0.37 | 63.43 ± 0.28 | 69.91 ± 0.18 |
| 10 | 85.11 ± 0.41 | 85.11 ± 0.41 | 62.88 ± 1.35 | 69.34 ± 1.20 | 66.82 ± 3.75 | 72.34 ± 3.04 |

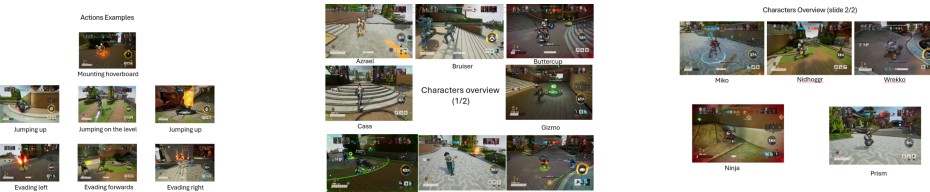

Figure 10: Reference slides shown to annotators during the human annotation study, illustrating the two recognition targets: *actions* (left) and *characters* (center and right). The slides include 20 exemplar videos (7 actions, 13 characters) to support consistent evaluation of VLM-generated responses.

**Annotation Setup and Interface.** Annotations were collected using a custom PowerPoint-based interface (see Figure 11). Each slide presented a short video, a question, and a generated answer. Annotators selected a rating from a predefined rubric. The full annotation guidelines – including action and character definitions and rating instructions – were embedded in the annotation deck for reference. For completeness, we also provide them in Table 13 and Figure 10. The annotation study was carried out by a subset of the authors with prior experience in the environment. Judging correctness required non-trivial familiarity with the visual dynamics and task ontology, making expert annotation necessary. All annotators were compensated above local minimum wage rates. Each QA pair was independently rated by two primary annotators. In cases of disagreement or if either

Table 8: *Supervision and Temporal Context:* Training budget and temporal context analysis for Action Recognition. Models are fine-tuned for up to 10 epochs and evaluated with up to 8 input frames.

| | | Binary | | Multiple-choice | | Open-ended | |
|---|---|---|---|---|---|---|---|
| Ep | Fr | EM | ROUGE | EM | ROUGE | EM | ROUGE |
| 1 | 1 | 83.97 ± 0.02 | 83.97 ± 0.02 | 61.43 ± 0.58 | 68.05 ± 0.70 | 61.68 ± 0.35 | 68.46 ± 0.19 |
| | 2 | 84.42 ± 0.06 | 84.42 ± 0.06 | 65.53 ± 0.27 | 72.03 ± 0.06 | 65.38 ± 0.06 | 71.74 ± 0.23 |
| | 4 | 90.97 ± 0.10 | 90.97 ± 0.10 | 83.11 ± 0.08 | 87.13 ± 0.04 | 82.26 ± 0.14 | 87.02 ± 0.06 |
| | 8 | 93.85 ± 0.28 | 93.85 ± 0.28 | 88.89 ± 0.14 | 93.40 ± 0.76 | 87.80 ± 0.20 | 92.23 ± 0.16 |
| 2 | 1 | 85.10 ± 0.02 | 85.10 ± 0.02 | 64.93 ± 0.06 | 71.09 ± 0.11 | 64.05 ± 0.01 | 70.36 ± 0.01 |
| | 2 | 86.53 ± 0.45 | 86.40 ± 0.26 | 69.20 ± 0.88 | 75.02 ± 0.67 | 68.83 ± 0.47 | 72.45 ± 2.76 |
| | 4 | 92.26 ± 0.34 | 92.26 ± 0.34 | 84.19 ± 0.10 | 88.46 ± 0.11 | 83.34 ± 0.15 | 87.84 ± 0.06 |
| | 8 | 95.05 ± 0.15 | 95.05 ± 0.15 | 89.27 ± 0.14 | 93.30 ± 0.22 | 89.42 ± 0.22 | 93.25 ± 0.15 |
| 4 | 1 | 85.37 ± 0.30 | 85.37 ± 0.30 | 64.58 ± 0.69 | 70.89 ± 0.55 | 64.79 ± 0.31 | 70.88 ± 0.27 |
| | 2 | 86.89 ± 0.06 | 86.89 ± 0.06 | 69.89 ± 0.83 | 75.49 ± 0.69 | 70.04 ± 0.08 | 75.74 ± 0.06 |
| | 4 | 92.58 ± 0.18 | 92.58 ± 0.18 | 85.13 ± 0.19 | 89.28 ± 0.17 | 84.61 ± 0.07 | 88.81 ± 0.04 |
| | 8 | 95.29 ± 0.07 | 95.29 ± 0.07 | 90.64 ± 0.00 | 94.09 ± 0.04 | 90.18 ± 0.16 | 93.81 ± 0.11 |
| 8 | 1 | 85.04 ± 0.00 | 85.04 ± 0.00 | 63.53 ± 0.35 | 69.95 ± 0.37 | 63.62 ± 0.00 | 70.03 ± 0.00 |
| | 2 | 87.27 ± 0.40 | 87.27 ± 0.40 | 69.84 ± 0.66 | 75.44 ± 0.45 | 70.11 ± 0.65 | 75.75 ± 0.52 |
| | 4 | 92.97 ± 0.49 | 92.97 ± 0.49 | 85.32 ± 0.08 | 89.27 ± 0.08 | 84.93 ± 0.21 | 89.05 ± 0.11 |
| | 8 | 95.48 ± 0.21 | 95.48 ± 0.21 | 90.71 ± 0.14 | 94.15 ± 0.13 | 91.02 ± 0.28 | 93.96 ± 0.71 |
| 10 | 1 | 85.40 ± 0.00 | 85.40 ± 0.00 | 62.88 ± 1.35 | 69.34 ± 1.20 | 66.82 ± 3.75 | 72.34 ± 3.04 |
| | 2 | 87.17 ± 0.22 | 87.17 ± 0.22 | 70.18 ± 0.00 | 75.64 ± 0.00 | 69.59 ± 0.20 | 75.37 ± 0.07 |
| | 4 | 92.96 ± 0.37 | 92.96 ± 0.37 | 85.02 ± 0.58 | 89.05 ± 0.49 | 84.71 ± 0.08 | 88.94 ± 0.06 |
| | 8 | 96.03 ± 0.05 | 96.03 ± 0.05 | 90.75 ± 0.04 | 94.22 ± 0.05 | 91.00 ± 0.11 | 94.33 ± 0.09 |

Table 9: Comparison of frame sampling strategies for Action Recognition. We evaluate first-$n$ vs. uniform-$n$ sampling across varying temporal context lengths ($n \in \{1, \ldots, 8\}$).

| | | Binary | | Multiple-choice | | Open-ended | |
|---|---|---|---|---|---|---|---|
| | Fr | EM | ROUGE | EM | ROUGE | EM | ROUGE |
| First-N | 1 | 83.97 ± 0.02 | 83.97 ± 0.02 | 61.43 ± 0.58 | 68.05 ± 0.70 | 61.68 ± 0.35 | 68.46 ± 0.19 |
| | 2 | 84.42 ± 0.06 | 84.42 ± 0.06 | 65.53 ± 0.27 | 72.03 ± 0.06 | 65.38 ± 0.06 | 71.74 ± 0.23 |
| | 3 | 87.93 ± 0.28 | 87.93 ± 0.28 | 75.73 ± 0.18 | 81.07 ± 0.06 | 74.68 ± 0.16 | 80.21 ± 0.08 |
| | 4 | 90.97 ± 0.10 | 90.97 ± 0.10 | 83.11 ± 0.08 | 87.13 ± 0.04 | 82.26 ± 0.14 | 87.02 ± 0.06 |
| | 5 | 92.00 ± 0.30 | 92.00 ± 0.30 | 85.46 ± 0.34 | 89.84 ± 0.18 | 85.10 ± 0.16 | 89.47 ± 0.10 |
| | 6 | 92.95 ± 0.30 | 92.95 ± 0.30 | 86.86 ± 0.08 | 91.13 ± 0.06 | 86.59 ± 0.39 | 90.82 ± 0.30 |
| | 7 | 93.31 ± 0.03 | 93.31 ± 0.03 | 87.95 ± 0.08 | 92.06 ± 0.06 | 87.58 ± 0.17 | 91.82 ± 0.08 |
| | 8 | 93.85 ± 0.28 | 93.85 ± 0.28 | 88.89 ± 0.14 | 93.40 ± 0.76 | 87.80 ± 0.20 | 92.23 ± 0.16 |
| Uniform-N | 1 | 83.97 ± 0.02 | 83.97 ± 0.02 | 61.43 ± 0.58 | 68.05 ± 0.70 | 61.68 ± 0.35 | 68.46 ± 0.19 |
| | 2 | 90.47 ± 0.62 | 90.47 ± 0.62 | 83.93 ± 0.04 | 88.36 ± 0.08 | 82.68 ± 0.19 | 87.33 ± 0.01 |
| | 3 | 93.59 ± 0.07 | 93.59 ± 0.07 | 88.90 ± 0.11 | 92.85 ± 0.10 | 88.49 ± 0.24 | 92.57 ± 0.04 |
| | 4 | 93.57 ± 0.39 | 93.57 ± 0.39 | 89.94 ± 0.04 | 93.65 ± 0.01 | 89.56 ± 0.42 | 93.49 ± 0.28 |
| | 5 | 94.25 ± 0.04 | 94.25 ± 0.04 | 89.99 ± 0.18 | 93.70 ± 0.13 | 89.72 ± 0.10 | 93.63 ± 0.23 |
| | 6 | 94.01 ± 0.57 | 94.01 ± 0.57 | 90.03 ± 0.04 | 93.73 ± 0.06 | 90.09 ± 0.18 | 93.88 ± 0.16 |
| | 7 | 93.96 ± 0.16 | 93.96 ± 0.16 | 90.34 ± 0.23 | 94.00 ± 0.10 | 89.94 ± 0.11 | 93.73 ± 0.10 |
| | 8 | 94.62 ± 0.48 | 94.62 ± 0.48 | 90.72 ± 0.12 | 94.30 ± 0.10 | 90.30 ± 0.04 | 94.01 ± 0.02 |

annotator marked the example as *Unclear*, a third, more experienced adjudicator reviewed the pair and assigned a final rating.

**Selected World Models.** For our study, we aim to evaluate rollouts generated across different model scales, training diversities, and output resolutions, while keeping the underlying architecture general enough to apply broadly. To this end, we select two autoregressive world models (Kanervisto et al., 2025). The autoregressive formulation offers a flexible and widely adopted framework, and is the basis for many state-of-the-art private world models.

Table 10: *Optimizing Data Mix for Unified Multi-Task Evaluation:* Performance tradeoffs under varying task-level allocation ratios for Action ($\alpha_{AR}$) vs. Character Recognition ($\alpha_{CR}$), with a fixed format distribution ($\beta = 1/3$ per format). Evaluated across binary, multiple-choice (MC), and open-ended (OE) formats.

| | | Binary | | Multiple-choice | | Open-ended | |
|---|---|---|---|---|---|---|---|
| $\alpha_{AR}$ | $\alpha_{CR}$ | EM | ROUGE | EM | ROUGE | EM | ROUGE |
| | | | | Action Recognition | | | |
| 0.20 | 0.80 | $84.13 \pm 1.66$ | $84.13 \pm 1.66$ | $26.10 \pm 0.43$ | $39.04 \pm 0.10$ | $27.03 \pm 0.21$ | $39.60 \pm 0.13$ |
| 0.40 | 0.60 | $88.11 \pm 1.44$ | $88.11 \pm 1.44$ | $28.13 \pm 1.15$ | $40.93 \pm 0.57$ | $29.17 \pm 0.40$ | $41.66 \pm 0.51$ |
| 0.50 | 0.50 | $88.59 \pm 1.41$ | $88.59 \pm 1.41$ | $29.10 \pm 0.65$ | $41.20 \pm 0.16$ | $29.66 \pm 0.54$ | $41.17 \pm 0.22$ |
| 0.60 | 0.40 | $90.80 \pm 0.04$ | $90.80 \pm 0.04$ | $30.44 \pm 0.03$ | $42.54 \pm 0.01$ | $30.55 \pm 0.49$ | $42.32 \pm 0.60$ |
| 0.80 | 0.20 | $91.23 \pm 0.91$ | $91.23 \pm 0.91$ | $84.06 \pm 1.26$ | $89.42 \pm 1.00$ | $30.88 \pm 0.63$ | $42.85 \pm 0.42$ |
| | | | | Character Recognition | | | |
| 0.20 | 0.80 | $98.57 \pm 0.47$ | $98.57 \pm 0.47$ | $98.95 \pm 0.16$ | $98.95 \pm 0.16$ | $98.94 \pm 0.22$ | $98.97 \pm 0.21$ |
| 0.40 | 0.60 | $98.51 \pm 0.53$ | $98.51 \pm 0.53$ | $98.77 \pm 0.16$ | $98.77 \pm 0.16$ | $98.98 \pm 0.06$ | $98.98 \pm 0.06$ |
| 0.50 | 0.50 | $96.33 \pm 1.81$ | $96.33 \pm 1.81$ | $98.03 \pm 0.06$ | $98.03 \pm 0.06$ | $98.23 \pm 0.06$ | $98.23 \pm 0.06$ |
| 0.60 | 0.40 | $93.22 \pm 3.38$ | $93.22 \pm 3.38$ | $96.91 \pm 1.81$ | $96.94 \pm 1.77$ | $97.93 \pm 0.02$ | $97.93 \pm 0.02$ |
| 0.80 | 0.20 | $80.53 \pm 0.49$ | $80.53 \pm 0.49$ | $89.08 \pm 0.49$ | $89.08 \pm 0.49$ | $89.02 \pm 0.25$ | $89.18 \pm 0.39$ |

Table 11: *Optimizing Data Mix for Unified Multi-Task Evaluation:* Performance on Action and Character Recognition under varying format-level sampling ratios ($\beta$) for Binary, Multiple-choice (MC), and Open-ended (OE) questions. We fix $\alpha_{AR} = 0.8$ and train all models on the first 8 frames.

| Ep | $\beta_{\texttt{binary}}$ | $\beta_{MC}$ | $\beta_{OE}$ | Binary EM | Binary ROUGE | Multiple-Choice EM | Multiple-Choice ROUGE | Open-Ended EM | Open-Ended ROUGE |
|---|---|---|---|---|---|---|---|---|---|
| | | | | | | Action Recognition | | | |
| 1 | 0.4 | 0.2 | 0.4 | $92.32 \pm 0.37$ | $92.32 \pm 0.37$ | $84.32 \pm 0.89$ | $89.54 \pm 0.64$ | $31.61 \pm 0.23$ | $43.51 \pm 0.09$ |
| | 0.2 | 0.4 | 0.4 | $90.80 \pm 0.71$ | $90.80 \pm 0.71$ | $86.60 \pm 0.38$ | $91.27 \pm 0.42$ | $32.06 \pm 0.18$ | $43.58 \pm 0.54$ |
| | 0.0 | 0.4 | 0.6 | $49.98 \pm 0.04$ | $49.98 \pm 0.04$ | $86.65 \pm 0.99$ | $91.26 \pm 0.87$ | $85.51 \pm 1.78$ | $90.13 \pm 1.53$ |
| | 0.0 | 0.2 | 0.8 | $50.11 \pm 0.06$ | $50.11 \pm 0.06$ | $86.58 \pm 0.47$ | $91.42 \pm 0.21$ | $87.45 \pm 0.09$ | $91.78 \pm 0.09$ |
| 2 | 0.4 | 0.2 | 0.4 | $93.14 \pm 0.48$ | $93.14 \pm 0.48$ | $86.77 \pm 0.00$ | $91.28 \pm 0.00$ | $32.96 \pm 0.00$ | $44.00 \pm 0.00$ |
| | 0.2 | 0.4 | 0.4 | $92.89 \pm 0.16$ | $92.89 \pm 0.16$ | $87.83 \pm 0.00$ | $92.13 \pm 0.00$ | $33.37 \pm 0.00$ | $44.13 \pm 0.00$ |
| | 0.0 | 0.4 | 0.6 | $41.22 \pm 0.00$ | $41.30 \pm 0.01$ | $89.17 \pm 0.07$ | $93.12 \pm 0.02$ | $88.55 \pm 0.00$ | $93.65 \pm 0.00$ |
| | 0.0 | 0.2 | 0.8 | $49.98 \pm 0.03$ | $49.99 \pm 0.02$ | $88.68 \pm 0.00$ | $92.71 \pm 0.00$ | $88.59 \pm 0.00$ | $92.56 \pm 0.00$ |
| 4 | 0.4 | 0.2 | 0.4 | $94.33 \pm 0.34$ | $94.33 \pm 0.34$ | $92.67 \pm 0.07$ | $93.27 \pm 0.71$ | $33.94 \pm 0.01$ | $43.96 \pm 0.00$ |
| | 0.2 | 0.4 | 0.4 | $94.19 \pm 0.13$ | $94.19 \pm 0.13$ | $93.04 \pm 0.01$ | $93.57 \pm 0.74$ | $33.78 \pm 0.00$ | $44.73 \pm 0.00$ |
| | 0.0 | 0.4 | 0.6 | $50.19 \pm 0.27$ | $50.19 \pm 0.27$ | $89.78 \pm 0.11$ | $93.52 \pm 0.03$ | $88.95 \pm 0.05$ | $92.75 \pm 0.06$ |
| | 0.0 | 0.2 | 0.8 | $49.98 \pm 0.02$ | $49.98 \pm 0.02$ | $89.25 \pm 0.00$ | $93.13 \pm 0.00$ | $89.41 \pm 0.00$ | $93.16 \pm 0.00$ |
| | | | | | | Character Recognition | | | |
| 1 | 0.4 | 0.2 | 0.4 | $86.42 \pm 0.25$ | $86.42 \pm 0.25$ | $94.77 \pm 0.14$ | $94.77 \pm 0.14$ | $94.76 \pm 0.08$ | $94.63 \pm 0.21$ |
| | 0.2 | 0.4 | 0.4 | $77.57 \pm 0.01$ | $77.57 \pm 0.01$ | $94.93 \pm 0.03$ | $94.93 \pm 0.03$ | $94.51 \pm 0.57$ | $94.15 \pm 0.00$ |
| | 0.0 | 0.4 | 0.6 | $50.37 \pm 0.52$ | $50.37 \pm 0.52$ | $96.56 \pm 0.28$ | $96.56 \pm 0.28$ | $96.81 \pm 0.39$ | $96.82 \pm 0.37$ |
| | 0.0 | 0.2 | 0.8 | $50.51 \pm 0.03$ | $50.51 \pm 0.03$ | $96.88 \pm 0.27$ | $96.88 \pm 0.27$ | $97.39 \pm 0.18$ | $97.39 \pm 0.18$ |
| 2 | 0.4 | 0.2 | 0.4 | $89.95 \pm 0.46$ | $89.95 \pm 0.46$ | $87.35 \pm 0.00$ | $87.36 \pm 0.00$ | $88.64 \pm 0.00$ | $88.64 \pm 0.00$ |
| | 0.2 | 0.4 | 0.4 | $91.28 \pm 0.20$ | $91.28 \pm 0.20$ | $93.90 \pm 0.00$ | $93.93 \pm 0.04$ | $93.06 \pm 0.00$ | $93.09 \pm 0.00$ |
| | 0.0 | 0.4 | 0.6 | $47.37 \pm 0.02$ | $47.48 \pm 0.04$ | $97.69 \pm 0.00$ | $97.70 \pm 0.00$ | $98.07 \pm 0.00$ | $98.07 \pm 0.00$ |
| | 0.0 | 0.2 | 0.8 | $50.07 \pm 0.04$ | $50.07 \pm 0.04$ | $97.75 \pm 0.00$ | $97.79 \pm 0.00$ | $98.07 \pm 0.00$ | $98.07 \pm 0.00$ |
| 4 | 0.4 | 0.2 | 0.4 | $97.71 \pm 0.05$ | $97.71 \pm 0.05$ | $97.70 \pm 0.00$ | $97.70 \pm 0.00$ | $97.88 \pm 0.01$ | $97.91 \pm 0.03$ |
| | 0.2 | 0.4 | 0.4 | $96.55 \pm 0.06$ | $96.55 \pm 0.06$ | $98.51 \pm 0.00$ | $98.51 \pm 0.00$ | $98.46 \pm 0.00$ | $98.46 \pm 0.00$ |
| | 0.0 | 0.4 | 0.6 | $51.25 \pm 1.56$ | $51.25 \pm 1.56$ | $98.21 \pm 0.13$ | $98.21 \pm 0.13$ | $98.48 \pm 0.06$ | $98.49 \pm 0.06$ |
| | 0.0 | 0.2 | 0.8 | $50.93 \pm 0.10$ | $50.93 \pm 0.10$ | $98.79 \pm 0.00$ | $98.79 \pm 0.00$ | $99.08 \pm 0.00$ | $99.09 \pm 0.00$ |

The selected models generate sequences of visual frames and controller actions without textual supervision, using a decoder-only transformer (Radford et al., 2019; Vaswani et al., 2017) trained autoregressively on discrete tokens. Visual frames are encoded with a VQGAN (Esser et al., 2021), while joystick actions are tokenized using a learned discretization scheme based on action bucketization (Kanervisto et al., 2020).

Table 12: *Optimizing Data Mix for Unified Multi-Task Evaluation:* Performance on Action and Character Recognition under high open-ended (OE) supervision, with $\beta_{\text{OE}} = 0.8$ and remaining budget split between Binary and Multiple-choice (MC).

| Ep | $\beta_{\text{BN}}$ | $\beta_{\text{MC}}$ | $\beta_{\text{OE}}$ | Binary | | Multiple-Choice | | Open-Ended | |
|---|---|---|---|---|---|---|---|---|---|
| | | | | EM | ROUGE | EM | ROUGE | EM | ROUGE |
| | | | | | | Action Recognition | | | |
| 1 | 0.15 | 0.05 | 0.80 | $88.85 \pm 0.04$ | $88.85 \pm 0.04$ | $81.93 \pm 0.00$ | $89.08 \pm 0.00$ | $86.72 \pm 0.00$ | $91.33 \pm 0.00$ |
| | 0.10 | 0.10 | 0.80 | $87.38 \pm 0.59$ | $87.38 \pm 0.59$ | $85.58 \pm 0.00$ | $90.50 \pm 0.00$ | $86.88 \pm 0.00$ | $91.25 \pm 0.00$ |
| | 0.05 | 0.15 | 0.80 | $85.67 \pm 0.19$ | $85.67 \pm 0.19$ | $86.38 \pm 0.09$ | $91.21 \pm 0.06$ | $86.84 \pm 0.02$ | $91.34 \pm 0.03$ |
| 2 | 0.15 | 0.05 | 0.80 | $92.45 \pm 0.11$ | $92.45 \pm 0.11$ | $87.52 \pm 0.00$ | $91.90 \pm 0.00$ | $87.97 \pm 0.63$ | $92.19 \pm 0.41$ |
| | 0.10 | 0.10 | 0.80 | $92.11 \pm 0.18$ | $92.11 \pm 0.18$ | $88.42 \pm 0.00$ | $92.54 \pm 0.00$ | $88.50 \pm 0.00$ | $92.54 \pm 0.00$ |
| | 0.05 | 0.15 | 0.80 | $91.98 \pm 0.24$ | $91.98 \pm 0.24$ | $88.72 \pm 0.00$ | $92.78 \pm 0.00$ | $88.56 \pm 0.00$ | $92.66 \pm 0.00$ |
| 4 | 0.15 | 0.05 | 0.80 | $92.98 \pm 0.21$ | $92.98 \pm 0.21$ | $88.93 \pm 0.00$ | $93.02 \pm 0.00$ | $89.64 \pm 0.00$ | $93.34 \pm 0.00$ |
| | 0.10 | 0.10 | 0.80 | $92.81 \pm 0.11$ | $92.81 \pm 0.11$ | $91.40 \pm 2.81$ | $93.88 \pm 0.70$ | $89.43 \pm 0.00$ | $93.20 \pm 0.00$ |
| | 0.05 | 0.15 | 0.80 | $91.52 \pm 0.37$ | $91.52 \pm 0.37$ | $89.80 \pm 0.00$ | $93.54 \pm 0.01$ | $89.81 \pm 0.01$ | $93.49 \pm 0.06$ |
| | | | | | | Character Recognition | | | |
| 1 | 0.15 | 0.05 | 0.80 | $59.75 \pm 0.04$ | $59.75 \pm 0.04$ | $95.45 \pm 0.00$ | $95.45 \pm 0.00$ | $97.16 \pm 0.00$ | $97.16 \pm 0.00$ |
| | 0.10 | 0.10 | 0.80 | $56.31 \pm 0.53$ | $56.31 \pm 0.53$ | $94.55 \pm 0.00$ | $94.55 \pm 0.00$ | $95.96 \pm 0.00$ | $95.96 \pm 0.00$ |
| | 0.05 | 0.15 | 0.80 | $50.86 \pm 0.08$ | $50.86 \pm 0.08$ | $96.87 \pm 0.02$ | $96.87 \pm 0.02$ | $97.12 \pm 0.01$ | $97.12 \pm 0.01$ |
| 2 | 0.15 | 0.05 | 0.80 | $80.18 \pm 0.09$ | $80.18 \pm 0.09$ | $95.41 \pm 0.00$ | $95.41 \pm 0.00$ | $96.91 \pm 0.00$ | $96.91 \pm 0.00$ |
| | 0.10 | 0.10 | 0.80 | $70.20 \pm 0.21$ | $70.20 \pm 0.21$ | $98.02 \pm 0.00$ | $98.02 \pm 0.00$ | $98.15 \pm 0.00$ | $98.15 \pm 0.00$ |
| | 0.05 | 0.15 | 0.80 | $69.67 \pm 0.36$ | $69.67 \pm 0.36$ | $97.37 \pm 0.00$ | $97.37 \pm 0.00$ | $97.79 \pm 0.00$ | $97.79 \pm 0.00$ |
| 4 | 0.15 | 0.05 | 0.80 | $94.16 \pm 0.12$ | $94.16 \pm 0.12$ | $97.06 \pm 0.00$ | $97.07 \pm 0.00$ | $97.91 \pm 0.00$ | $97.91 \pm 0.00$ |
| | 0.10 | 0.10 | 0.80 | $86.67 \pm 0.13$ | $86.67 \pm 0.13$ | $98.50 \pm 0.00$ | $98.50 \pm 0.00$ | $98.77 \pm 0.00$ | $98.77 \pm 0.00$ |
| | 0.05 | 0.15 | 0.80 | $71.79 \pm 0.01$ | $71.79 \pm 0.01$ | $98.22 \pm 0.00$ | $98.22 \pm 0.00$ | $98.57 \pm 0.00$ | $98.57 \pm 0.00$ |

Each world model is implemented as a decoder-only transformer (Radford et al., 2019; Vaswani et al., 2017), trained to predict discrete tokens representing visual observations and actions. Visual frames are first compressed with a VQGAN (Esser et al., 2021), while joystick actions are tokenized using a learned discretization scheme based on action bucketization (Kanervisto et al., 2020). The objective is next-token prediction conditioned on prior visual and action tokens. We focus on the following model variants that differ in capacity, training diversity, and output resolution: (i) a large-scale model: 1.6B parameters, trained for 200K steps on gameplay from seven diverse environments (including Skygarden) at $300 \times 180$, and (ii) a smaller model: 140M parameters, trained for 100K steps on gameplay from a single environment (Skygarden) at $128 \times 128$ resolution.

**Rollouts Generation.** Rollout generation follows a consistent protocol for both world models: at inference time, the model is conditioned on 1 second of ground-truth gameplay (visual and action tokens), after which it generates 10 seconds of future gameplay conditioned only on a sequence of held-out controller actions. For rollout generation, we use reference trajectories from the game dataset, which include time-aligned controller inputs and image frames. For each generated rollout, we randomly sample a 1-second context window (video frames and corresponding action tokens) from these trajectories. The world model is then conditioned on this context, which consists of an interleaved sequence of tokenized images and actions, and proceeds autoregressively to generate latent image tokens and discretized action tokens. The generated rollout is then split into 14-frame chunks. This setup enables a comprehensive analysis of the UNIVERSE's evaluation capabilities across two axes: (i) *in-domain performance*: evaluating on Skygarden, the environment used for fine-tuning; (ii) *generalization*: assessing performance on six unseen environments. It also allows comparison across generation quality and model capacity. We generate 82 rollouts for each model-environment setting, resulting in 656 rollouts in total.

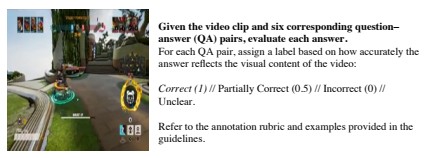

Figure 11: Annotation interface example. Each instance includes a video clip, task instructions, and a table with: *Question* (generated via evaluation protocol), *Response* (VLM output), and *Score* (human-assigned label).

Table 13: Annotation instructions provided to human raters as part of the study. The interface outlines task context, scoring criteria, general guidelines, and reference definitions for supported action categories.

| **1. Task Overview** |
| --- |
| You will be presented with:
• A short video clip;
• A natural language question about the video;
• An answer generated by a vision-language model.
Your task is to evaluate whether the model's answer accurately describes the events depicted in the video. |
| **2. How to Rate Each Answer** |
| Assign one of the following categories:
• *Correct (1.0)*: Fully matches the event in the video;
• *Partially Correct (0.5)*: Captures the general idea but contains a minor error;
• *Incorrect (0.0)*: Wrong, hallucinated, or mismatched with the visual evidence;
• *Unclear / Cannot Tell*: Not enough evidence to confidently decide. |
| **3. General Guidelines** |
| • Watch the full video before rating;
• Base your decision solely on visible content;
• Use provided action and character references;
• If multiple plausible interpretations exist and the answer matches one, mark as *Correct*;
• If unsure even after review, mark *Unclear / Cannot Tell*;
• Optionally leave comments for ambiguous or interesting cases. |
| **5. Action Label Definitions** |
| • *Evading Backwards*: Moves backwards to avoid threat or reposition.
• *Evading Forwards*: Moves forwards.
• *Evading Left / Right*: Lateral movement left or right.
• *Jumping Down*: Jumps from a higher to a lower platform or level.
• *Jumping on the Level*: Jumps without elevation change.
• *Jumping Up*: Jumps upward to reach a higher platform.
• *Mounting Hoverboard*: Begins riding or is seen riding a hoverboard. |

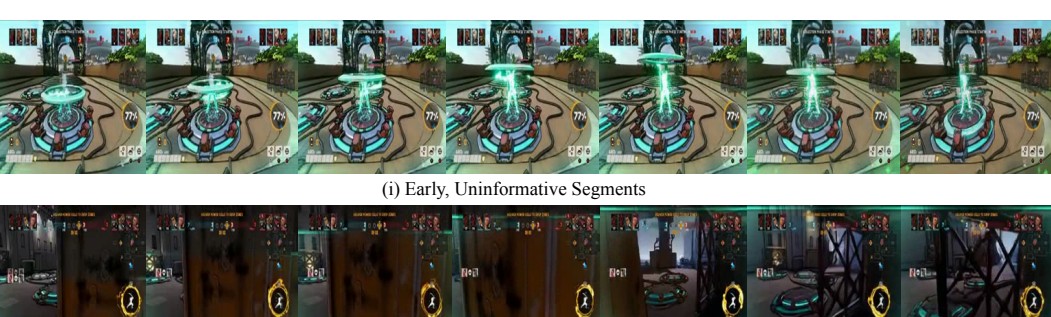

(i) Early, Uninformative Segments

(ii) Rollouts Dominated by Occlusion

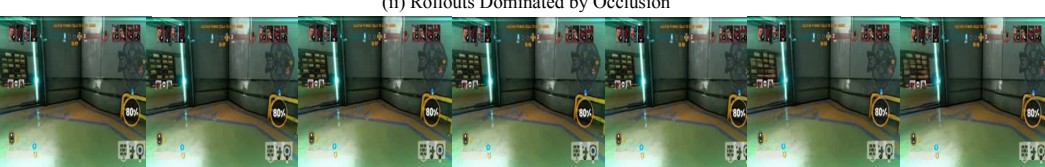

(iii) Sequence with No Visible Agents

Figure 12: Randomly sampled examples of rollouts excluded from the human evaluation study.

**Rollout Filtering.** To keep evaluation focused on challenging sequences, we remove only those

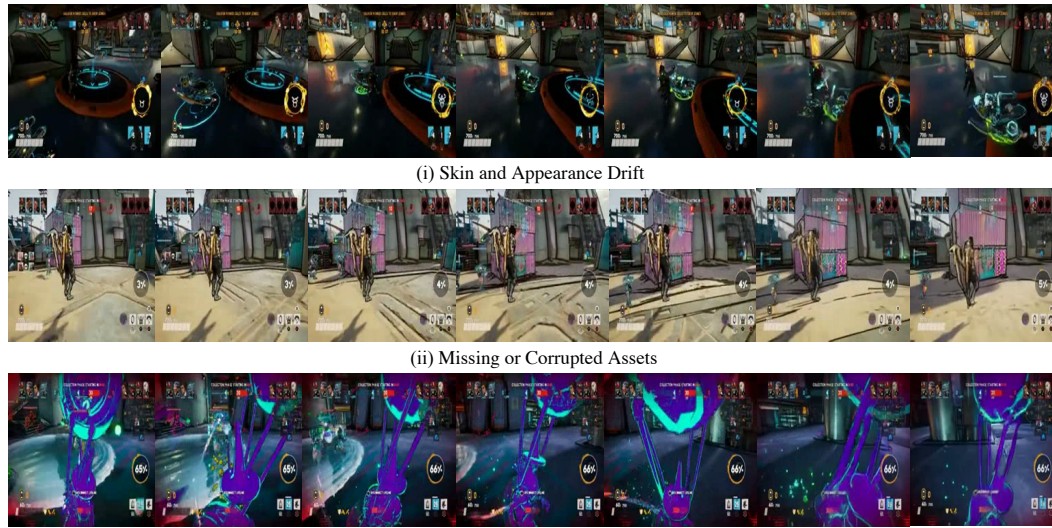

(i) Skin and Appearance Drift

(ii) Missing or Corrupted Assets

(iii) Heads-Up Display-Induced Occlusion

Figure 13: Randomly sampled examples of rollouts retained for human evaluation.

rollouts that are either uninterpretable or uninformative. Specifically, annotators discard: (i) early segments with no meaningful interactions; (ii) rollouts dominated by occlusion; and (iii) sequences with no visible agents. These criteria target cases where neither humans nor models can extract actionable dynamics, while retaining all visually imperfect, ambiguous, and failure-prone generations. Thus, filtering eliminates only sequences that provide no evaluable signal. Figure 12 shows randomly sampled examples from each excluded category.

**Complexity of Retained Rollouts.** Despite filtering out uninterpretable/uninformative sequences, many retained rollouts exhibit visual imperfections that make evaluation challenging. Annotators highlighted recurring generation failures including: (i) skin and appearance drift, where model-generated agent skins diverge from canonical silhouettes; (ii) missing or corrupted assets, such as hoverboards implied only through motion cues; and (iii) HUD-induced occlusion, where oversized or misplaced UI elements block important scene content. These imperfections arise from the underlying world models and introduce ambiguity while still permitting meaningful assessment. Figure 13 illustrates representative failure-prone cases, emphasizing that UNIVERSE is evaluated on realistic, imperfect rollouts rather than sanitized inputs.

**UNIVERSE's Response Generation.** To obtain responses from UNIVERSE, we provide it with a video segment (resized to match the evaluator's input resolution) along with its corresponding question. We then sample five responses using greedy decoding and select the most frequent response as the final answer. In cases where all five responses are unique (i.e., no majority), one response is selected at random. The resulting dataset comprises rollouts from 8 settings: rollouts generated by a smaller model on Skygarden, and rollouts generated by a larger model across seven distinct environments. For each model–environment pair, we sample 30 rollouts. Each rollout is annotated with 6 question–answer (QA) pairs, along with a corresponding response from the adapted evaluator. Each of the resulting 1,440 QA instances was rated by 3 annotators, yielding 4,320 total human judgments.

F.2   EVALUATION METRICS

We report two accuracy-based metrics using the adjudicated labels:

*Strict Accuracy.*: The proportion of QA pairs labeled as *Correct*:

$$\text{Acc}_{\text{Strict}} = \frac{N_{\text{Correct}}}{N_{\text{Answerable}}}, \tag{5}$$

Table 14: Inter-annotator agreement and valid QA coverage across environments. We report Cohen's $\kappa$ between the two primary annotators for each world model–map pair. The total number of valid examples excludes QA pairs marked as *Unclear* by at least one annotator.

| Setting | Valid QA Pairs | Cohen's $\kappa$ |
|---------|----------------|------------------|
| 1 | 29 | 0.91 |
| 2 | 28 | 0.67 |
| 3 | 28 | 0.74 |
| 4 | 29 | 0.87 |
| 5 | 30 | 0.67 |
| 6 | 29 | 0.61 |
| 7 | 30 | 0.59 |
| 8 | 24 | 0.79 |

Table 15: Graded/strict accuracy of UNIVERSE on Action and Character Recognition tasks, evaluated by human annotators across different environments and question formats. We report results for Binary, Multiple-Choice (MC), and Open-Ended (OE) prompts, disaggregated by task and world model. All metrics are based on final adjudicated ratings.

| Setting | Binary | Action Recognition | | | Character Recognition | |
| | | MC | OE | Binary | MC | OE |
|---------|--------|------|------|--------|------|------|
| 1 | 98.3 / 96.7 | 51.7 / 46.7 | 75.0 / 73.3 | 93.3 / 93.3 | 83.3 / 83.3 | 93.3 / 93.3 |
| 2 | 96.7 / 96.7 | 60.0 / 60.0 | 65.0 / 60.0 | 99.9 / 99.9 | 90.0 / 90.0 | 93.3 / 93.3 |
| 3 | 96.7 / 96.7 | 63.3 / 63.3 | 80.0 / 80.0 | 99.9 / 99.9 | 86.7 / 86.7 | 93.3 / 93.3 |
| 4 | 93.3 / 93.3 | 43.3 / 43.3 | 73.3 / 73.3 | 96.7 / 96.7 | 96.7 / 96.7 | 99.9 / 99.9 |
| 5 | 80.0 / 76.7 | 76.7 / 73.3 | 93.3 / 93.3 | 96.7 / 96.7 | 99.9 / 99.9 | 99.8 / 99.8 |
| 6 | 71.7 / 70.0 | 56.7 / 56.7 | 75.0 / 70.0 | 96.7 / 96.7 | 93.3 / 93.3 | 96.7 / 96.7 |
| 7 | 68.3 / 66.7 | 50.0 / 46.7 | 80.0 / 76.7 | 93.3 / 93.3 | 90.0 / 90.0 | 96.7 / 96.7 |
| 8 | 92.9 / 89.3 | 35.7 / 32.1 | 41.1 / 39.3 | 85.7 / 85.7 | 10.7 / 10.7 | 60.7 / 60.7 |

*Graded Accuracy.*: Partial credit given to *Partially Correct* responses:

$$\text{Acc}_{\text{Graded}} = \frac{N_{\text{Correct}} + 0.5 \times N_{\text{Partial}}}{N_{\text{Answerable}}}. \tag{6}$$

Only examples not marked *Unclear* by adjudication are included in $N_{\text{Answerable}}$.

*Inter-Annotator Agreement.* To quantify rating consistency, we compute Cohen's $\kappa$ between the two primary annotators. The adjudicator's label is used only when disagreement occurs and is excluded from agreement computation. Results are shown in Table 14.

*Sample Size Justification.* We annotate 30 rollouts per model–environment pair. Assuming a standard deviation of $\sigma \approx 0.2$ and a 95% confidence level, the confidence interval (CI) width is given by CI Width $= z_{\frac{1-C}{2}} \cdot \frac{\sigma}{\sqrt{n}}$. This yields an estimated CI of $\sim$7.1% for individual model–environment pairs ($n = 30$), and $\sim$2.5% when aggregating across all eight pairs ($n = 240$), offering sufficient precision for comparative evaluation.

### F.3 RESULTS

Table 15 reports graded and strict accuracy across environments, recognition targets (Action and Character Recognition), and question formats (Binary, Multiple-Choice, Open-Ended). We observe a clear gap in performance between rollouts generated by the two world models. UNIVERSE struggles with outputs from WHAM 140M, achieving substantially lower accuracy compared to WHAM 1.6B. This is likely due to a mismatch in image resolution: WHAM 140M generates frames at $128 \times 128$ resolution, which must be upsampled to the UNIVERSE's expected input of $224 \times 224$. Despite resizing, the resulting frames often lack sharpness, making actions and characters harder to recognize. In

contrast, UNIVERSE performs well on rollouts from WHAM 1.6B, even across diverse environments. On the in-domain setting (Environment A), the model achieves strong results—averaging 75.02% graded accuracy for AR and 90.00% for CR. When evaluating on the six unseen environments (Environments B–G), performance for AR drops slightly (from 75.02% to 73.52%), while CR remains stable or improves, suggesting strong generalization in character grounding and visual consistency tracking.

**Qualitative Examples.** Figure 14 illustrates the diversity of generated rollouts across environments. WHAM 1.6B captures greater visual variation and scene composition compared to WHAM 140M.

# G    SUPPLEMENTARY EXPERIMENTAL RESULTS

This section presents additional experimental results that support the main findings but are omitted from the main paper for clarity and space. These include: (i) a zero-shot analysis of PaliGemma variants to motivate backbone selection, (ii) CLIPScore-based baselines to contextualize performance without adaptation, and (iii) a study of low-rank adaptation (LoRA) across different rank values. While these results are not central to the unified evaluation framework proposed in the main text, they provide valuable insight into model selection, adaptation efficiency, and the limitations of standard evaluation proxies in our setting.

## G.1    GPT-5 PERFORMANCE

To demonstrate the complexity of the tasks that comprise our protocol, we conducted an evaluation of GPT-5 on randomly selected examples. We deliberately chose the simplest evaluation regime (binary action recognition) to test whether the model can succeed without adaptation.

Figure 15 demonstrates that out of six random samples, GPT-5 produced incorrect answers in five cases. This consistent failure highlights the difficulty of the task.

## G.2    ZERO-SHOT PERFORMANCE OF PALIGEMMA MODELS

In this section, we benchmark three pretrained configurations—PaliGemma 1 3b, PaliGemma 2 3b, and PaliGemma 2 10b—under our proposed protocol and motivate our choice of PaliGemma 2 3b as the default backbone for subsequent experiments. Each model receives a natural language prompt along with either 1 or 8 image frames as input and produces a textual response. This experiment probes both model capacity and the role of temporal visual context in zero-shot settings.

**Results.** Figure 16 reports ROUGE scores across task types, question formats, and visual context lengths. While zero-shot performance reveals some capacity for structured reasoning—particularly in the multiple-choice setting—it remains limited overall. Binary accuracy hovers near chance, and open-ended responses frequently lack specificity. Performance is strongest on action recognition (AR), likely reflecting pretrained models' familiarity with generic visual dynamics. In contrast, character recognition (CR) lags behind, underscoring a lack of grounding in domain-specific entities. Increasing the number of input frames modestly improves AR, but yields diminishing returns for CR. Among the evaluated configurations, PaliGemma 2 10b performs best in absolute terms. However, the margin over PaliGemma 2 3b is narrow, and PaliGemma 2 3b offers a substantially smaller footprint while using a newer Gemma 2 decoder architecture. We therefore adopt PaliGemma 2 3b as the default model for all subsequent adaptation experiments, balancing performance, compute efficiency, and architectural recency.

## G.3    CLIPSCORE COMPARISONS

To further evaluate zero-shot recognition capabilities without adaptation, we apply CLIPScore to our rollout evaluation protocol. Specifically, we assess four pretrained CLIP variants – ViT-B/32, ViT-B/16, ViT-L/14, and ViT-L/14-336 – across both Action Recognition (AR) and Character Recognition (CR) tasks using 1-frame and 8-frame visual inputs. For each evaluation instance, we extract either 1 or 8 frames from the video segment and compute the cosine similarity between each image and a predefined set of textual labels (i.e., action verbs for AR, character names for CR). For single-frame settings, we select the label with the highest similarity score as the predicted class. In the multi-frame

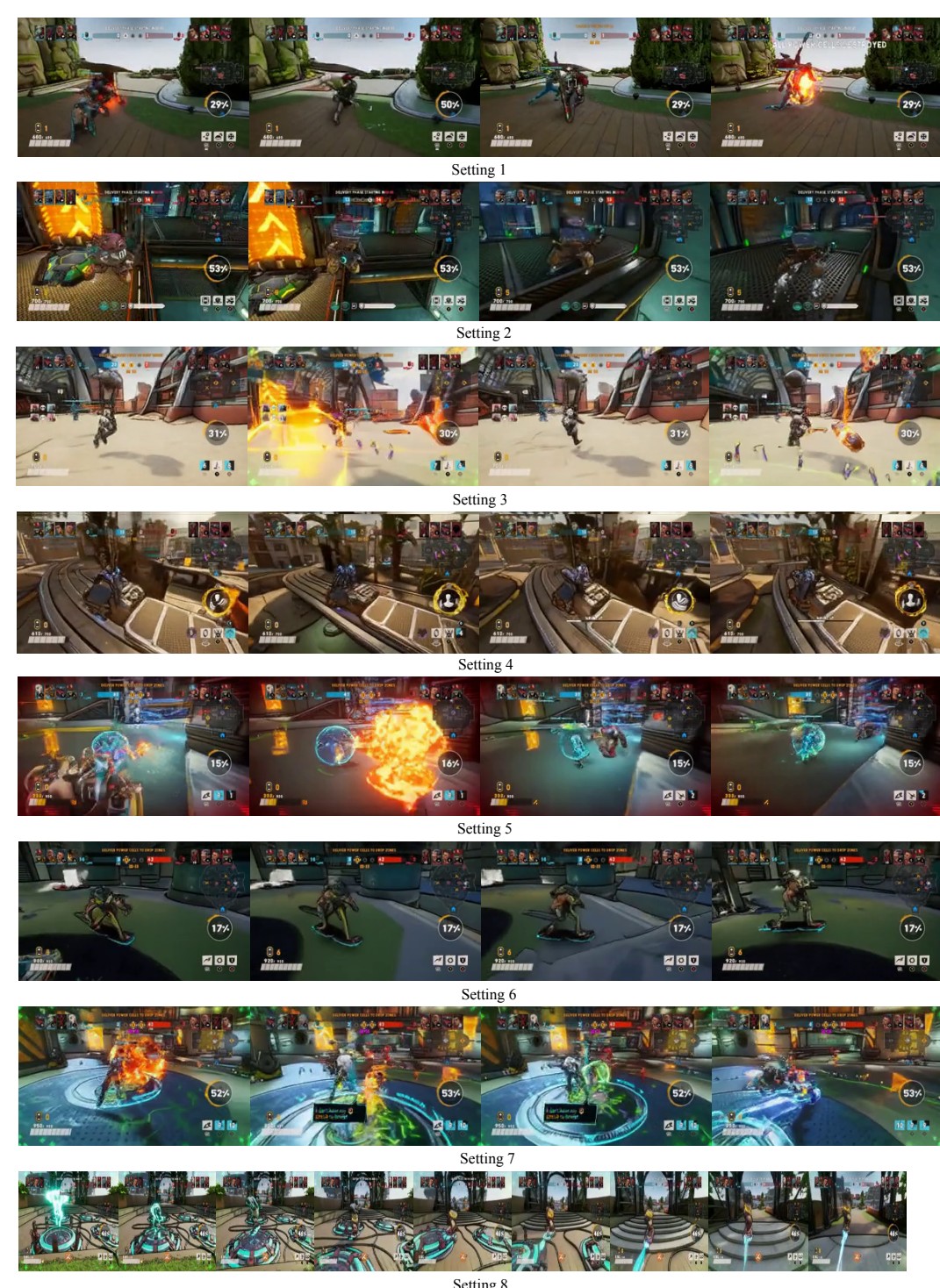

Figure 14: Representative frames from rollouts across the eight evaluation settings, spanning different environments, scales, and resolutions.

setting, we compute predictions for each frame independently and use a majority vote to produce the final prediction. We also report two reference baselines for context: a random classifier, which achieves 12.5% on AR and 7.7% on CR, and a majority-class predictor, which yields 35.5% and 17.6% respectively. These are included only for calibration.

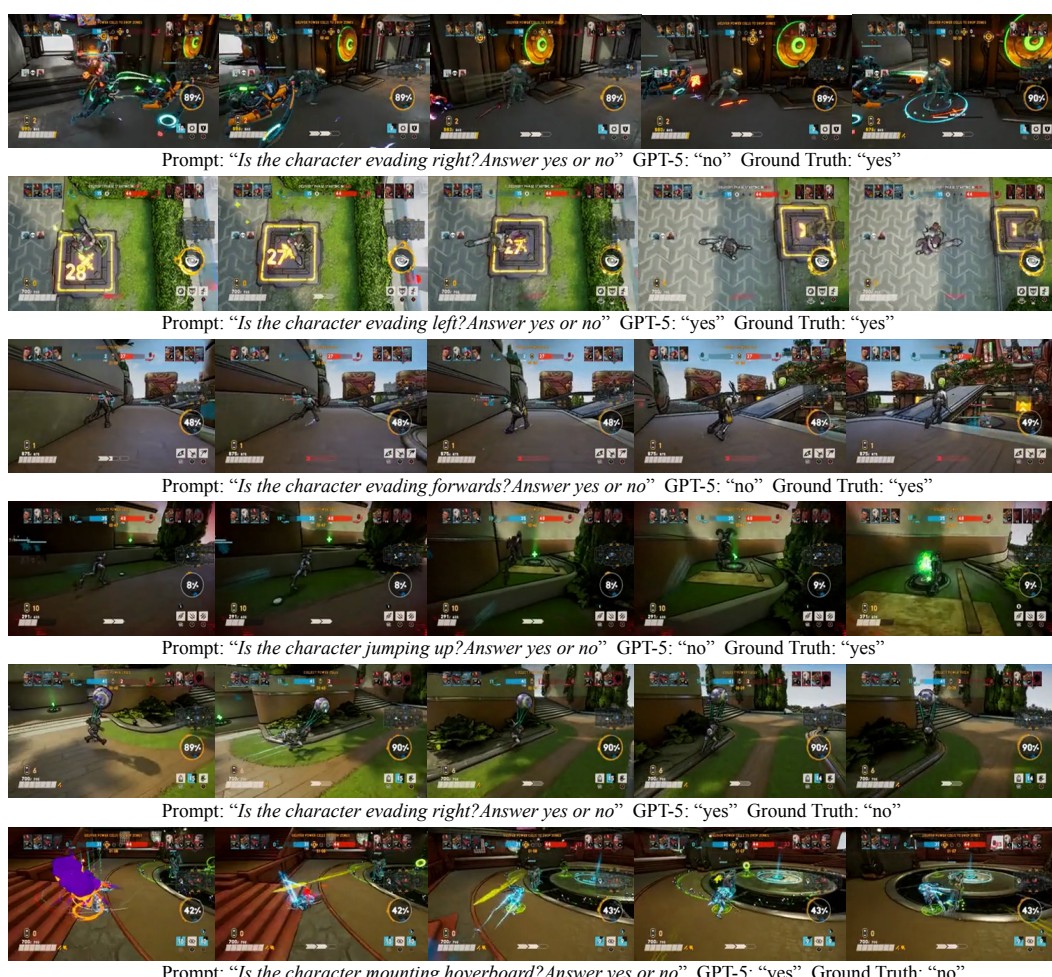

Prompt: "*Is the character evading right? Answer yes or no*" GPT-5: "no" Ground Truth: "yes"

Prompt: "*Is the character evading left? Answer yes or no*" GPT-5: "yes" Ground Truth: "yes"

Prompt: "*Is the character evading forwards? Answer yes or no*" GPT-5: "no" Ground Truth: "yes"

Prompt: "*Is the character jumping up? Answer yes or no*" GPT-5: "no" Ground Truth: "yes"

Prompt: "*Is the character evading right? Answer yes or no*" GPT-5: "yes" Ground Truth: "no"

Prompt: "*Is the character mounting hoverboard? Answer yes or no*" GPT-5: "yes" Ground Truth: "no"

Figure 15: Performance of GPT-5 on six randomly sampled binary action recognition questions. Each panel shows the trial prompt, the model's response, and the ground-truth label. Despite the apparent simplicity of the setup, GPT-5 fails on 5/6 trials, underscoring the difficulty of the task and the need for task-specific adaptation.

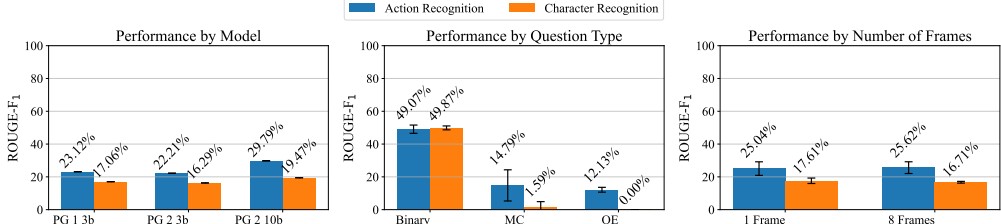

Figure 16: Zero-shot evaluation results for PaliGemma variants across tasks, prompt formats, and visual context sizes. Overall performance remains limited, indicating the need for task-specific adaptation.

**Results.** Table 16 demonstrates the results. While CLIP ViT-B/16 performs relatively well on AR in both input settings, performance remains inconsistent across model scales and tasks. In particular, CR accuracy remains low, reflecting CLIP's limited grounding in domain-specific visual semantics and fine-grained identity resolution. Larger CLIP models such as ViT-L/14 do not consistently outperform smaller variants, and 8-frame inputs provide only marginal gains over single-frame inputs.

Table 16: Zero-shot accuracy-based evaluation of CLIP models and baseline methods on Action and Character Recognition tasks using 1 and 8 input frames.

| Fr | Model | Action Recognition | Character Recognition |
|---|---|---|---|
| 1 | CLIP ViT-B/32 | $24.04 \pm 0.00$ | $13.32 \pm 0.00$ |
| | CLIP ViT-B/16 | $52.67 \pm 0.00$ | $16.47 \pm 0.00$ |
| | CLIP ViT-L/14 | $24.60 \pm 0.00$ | $9.95 \pm 0.00$ |
| | CLIP ViT-L/14-336 | $12.17 \pm 0.00$ | $8.85 \pm 0.05$ |
| 8 | CLIP ViT-B/32 | $36.22 \pm 0.00$ | $14.41 \pm 0.00$ |
| | CLIP ViT-B/16 | $57.36 \pm 0.00$ | $17.24 \pm 0.00$ |
| | CLIP ViT-L/14 | $17.57 \pm 0.00$ | $10.10 \pm 0.00$ |
| | CLIP ViT-L/14-336 | $23.12 \pm 0.00$ | $8.64 \pm 0.00$ |

Table 17: Performance on Action and Character Recognition tasks after LoRA-based adaptation with varying ranks ($r \in \{8, 16, 32, 48, 64\}$). Adapters are applied to attention and MLP layers in both vision and language components.

| | Binary | | Multiple-choice | | Open-ended | |
|---|---|---|---|---|---|---|
| Rank | EM | ROUGE | EM | ROUGE | EM | ROUGE |
| **Action Recognition** | | | | | | |
| 8 | $44.66 \pm 0.21$ | $44.66 \pm 0.21$ | $0.02 \pm 0.00$ | $9.21 \pm 0.00$ | $0.00 \pm 0.00$ | $12.49 \pm 0.00$ |
| 16 | $44.47 \pm 0.43$ | $44.47 \pm 0.43$ | $0.02 \pm 0.00$ | $9.21 \pm 0.00$ | $0.00 \pm 0.00$ | $12.49 \pm 0.00$ |
| 32 | $44.59 \pm 0.03$ | $44.59 \pm 0.03$ | $0.02 \pm 0.00$ | $9.21 \pm 0.00$ | $0.00 \pm 0.00$ | $12.49 \pm 0.00$ |
| 48 | $46.71 \pm 3.20$ | $46.71 \pm 3.20$ | $0.02 \pm 0.00$ | $9.21 \pm 0.00$ | $0.00 \pm 0.00$ | $12.49 \pm 0.00$ |
| 64 | $48.67 \pm 0.13$ | $48.67 \pm 0.13$ | $0.02 \pm 0.00$ | $9.21 \pm 0.00$ | $0.00 \pm 0.00$ | $12.49 \pm 0.00$ |
| **Character Recognition** | | | | | | |
| 8 | $48.76 \pm 0.00$ | $48.76 \pm 0.00$ | $0.00 \pm 0.00$ | $0.32 \pm 0.00$ | $0.00 \pm 0.00$ | $0.01 \pm 0.01$ |
| 16 | $48.62 \pm 0.23$ | $48.62 \pm 0.23$ | $0.00 \pm 0.00$ | $0.14 \pm 0.00$ | $0.00 \pm 0.00$ | $0.05 \pm 0.00$ |
| 32 | $48.98 \pm 0.08$ | $48.98 \pm 0.08$ | $0.00 \pm 0.00$ | $0.14 \pm 0.00$ | $0.00 \pm 0.00$ | $0.05 \pm 0.00$ |
| 48 | $48.91 \pm 0.09$ | $48.91 \pm 0.09$ | $0.00 \pm 0.00$ | $0.14 \pm 0.00$ | $0.00 \pm 0.00$ | $0.05 \pm 0.00$ |
| 64 | $48.72 \pm 0.06$ | $48.72 \pm 0.06$ | $0.00 \pm 0.00$ | $0.14 \pm 0.00$ | $0.00 \pm 0.00$ | $0.05 \pm 0.00$ |

Overall, these results suggest that while CLIPScore offers a lightweight and scalable evaluation proxy, it lacks the temporal grounding and semantic specificity required for structured rollout evaluation. Performance falls short relative to our selected baselines, and the method is inherently constrained to predefined candidate sets—limiting its applicability to open-ended or compositional tasks. As such, we exclude CLIP-based scores from our primary comparisons and instead focus on adapted, generative VLM-based evaluators.

### G.4 LOW-RANK ADAPTATION COMPARISONS

This section presents an extended analysis of low-rank adaptation (LoRA) as a parameter-efficient strategy for adapting vision-language models to our protocol. We systematically vary the rank parameter $r$ and measure its impact on Action and Character Recognition performance across all prompt formats. All experiments in this section are conducted using PaliGemma 2 (3B) as the backbone model, consistent with the main fine-tuning results. These experiments assess whether increasing rank provides meaningful gains, and inform our decision to report only the rank-8 setting in the main paper.

**Results.** Table 17 presents the performance of LoRA-based adaptation across a range of rank values ($r \in \{8, 16, 32, 48, 64\}$) for both Action Recognition (AR) and Character Recognition (CR) tasks, across all prompt formats. We report exact match (EM) and ROUGE-$F_1$ averaged over three runs. Increasing the rank beyond $r = 8$ yields no consistent improvements across tasks or formats. Performance on binary prompts remains close to random, while performance on multiple-choice and

Table 18: Pearson correlation between FVD and UNIVERSE across experimental settings. None of the correlations are statistically significant (all $p > 0.05$).

| Setting | Action Recognition | Character Recognition |
|---------|-------------------|----------------------|
| 1 | 0.09 | 0.03 |
| 2 | -0.07 | -0.08 |
| 3 | -0.17 | 0.06 |
| 4 | -0.03 | -0.10 |
| 5 | 0.07 | -0.25 |
| 6 | -0.01 | 0.21 |
| 7 | 0.32 | 0.03 |
| 8 | -0.10 | 0.33 |

open-ended formats stays near zero across all ranks. These results suggest that LoRA, even with increased capacity, is insufficient for capturing the fine-grained temporal and semantic dependencies required by our evaluation protocol. Given the lack of benefit from increasing rank—and the added parameter cost—it is inefficient to scale LoRA rank beyond $r = 8$. Accordingly, all results reported in the main paper use $r = 8$, while extended comparisons with higher ranks are presented here for completeness.

## H  COMPARISON WITH EXISTING EVALUATION BASELINES

To validate that UNIVERSE captures evaluation dimensions beyond existing automated metrics, we analyze its correlation with two representative baselines: Fréchet Video Distance (FVD), a standard distributional metric for video generation quality, and VBench, a comprehensive multi-dimensional benchmark for text-to-video evaluation. These comparisons assess whether UNIVERSE's focus on action-conditioned semantic alignment provides complementary signal to metrics designed primarily for perceptual quality and temporal coherence.

### H.1  COMPARISON WITH FVD

FVD Unterthiner et al. (2018) is a widely used reference-free metric for evaluating generative video models, defined as a distance between feature distributions of real and generated videos. Although commonly treated as a default quantitative proxy for rollout quality in world-model evaluation, FVD is agnostic to task semantics and action alignment—the dimensions UNIVERSE is specifically designed to capture. Across the eight environments, raw FVD values are tightly clustered (means between 3.23 and 3.27 with standard deviations below 0.05), reflecting limited dynamic range and strong dataset-dependent scale effects. As a result, raw scores are not directly comparable across settings, making correlations a more informative basis for analysis. We therefore assess the relationship between the two metrics by computing Pearson correlations between FVD and UNIVERSE's human-aligned scores across all environments for both Action Recognition and Character Recognition tasks.

**Results.** For each environment (Settings 1–8) and each task, we obtain FVD scores on the same set of rollouts evaluated by UNIVERSE, report the scores and compute the correlation between the two. The resulting coefficients are reported in Table 18. Correlations are small in magnitude and unstable in sign, ranging from $-0.25$ to $0.33$, with most values close to zero. The largest positive correlations are modest (Action Recognition, Setting 7: $r = 0.32$; Character Recognition, Setting 8: $r = 0.33$), and several settings exhibit weak negative correlations (e.g., Character Recognition, Setting 5: $r = -0.25$). None of these correlations are statistically significant ($p > 0.05$ for all settings), indicating UNIVERSE captures complementary semantic cues. These results demonstrate that FVD and UNIVERSE provide complementary evaluation signals: FVD captures distribution-level perceptual quality and temporal coherence, while UNIVERSE quantifies task-specific semantic alignment through human-grounded assessment.

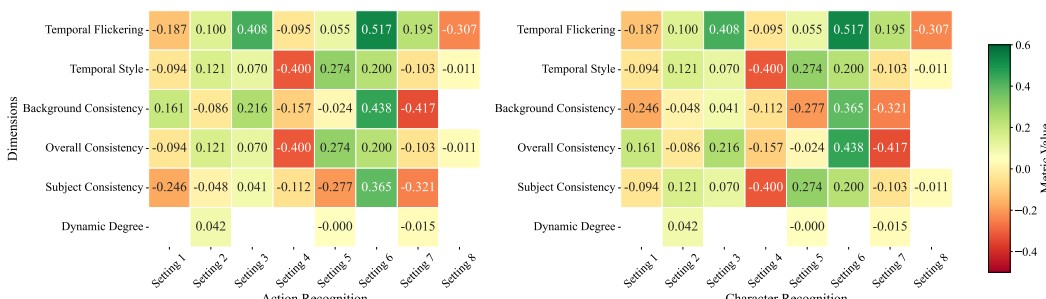

Figure 17: Correlation between UNIVERSE and VBench evaluation suite across eight environments and six dimensions. The *human action* dimension is omitted as it was constant and non-informative. Correlations remain consistently low to moderate ($|r| < 0.4$), with none of the correlations are statistically significant (all $p > 0.05$).

## H.2 CORRELATION WITH VBENCH METRICS

To validate that UNIVERSE captures evaluative dimensions beyond existing automated benchmarks, we compare its human-aligned evaluation scores against VBench Huang et al. (2024), a comprehensive state-of-the-art benchmark for generative video evaluation. We computed correlations on identical rollout sets across eight experimental environments using six evaluation configurations, parameterized by task (Action Recognition or Character Recognition) and prompt type (binary, multiple-choice, open-ended). The analysis spans seven VBench dimensions: Temporal Flickering, Temporal Style, Background Consistency, Overall Consistency, Subject Consistency, Dynamic Degree, and Human Action. The Human Action dimension is omitted from visualization as it exhibited no variance across samples.

**Results.** Figure 17 shows that correlations between UNIVERSE and VBench metrics remain consistently low to moderate ($|r| < 0.4$) across all valid comparisons. Average correlations range from 0.037 for Subject Consistency to 0.06 for Temporal Flickering, with most values falling within 0.1–0.4. Notably, the sign variability across settings (including both positive and negative correlations) indicates that UNIVERSE captures complementary semantic cues not reflected in VBench's quality dimensions. This divergence is expected: while VBench focuses on low-level perceptual quality and temporal coherence, UNIVERSE explicitly targets semantic alignment.

**Summary.** The consistent orthogonality observed across both analyses demonstrates that UNIVERSE provides a complementary evaluation dimension to existing benchmarks. While standard distributional metrics (FVD) capture visual fidelity and text-to-video benchmarks (VBench) assess perceptual quality, UNIVERSE quantifies action-conditioned semantic alignment through human-grounded evaluation. Comprehensive world model assessment therefore benefits from combining these approaches: existing metrics for perceptual coherence and UNIVERSE for task-relevant semantic fidelity.

