# OpenReview forum: "Adapting Vision-Language Models for Evaluating World Models"
_ICLR.cc/2026/Conference — Submitted to ICLR 2026_

### Official Review · Reviewer_dQyk · 2025-10-19

**Soundness:** 1
**Presentation:** 3
**Contribution:** 2
**Rating:** 2
**Confidence:** 4

**Summary:**

The paper proposes UNIVERSE, a new VLM-based method for the evaluation of "action recognition" and "character recognition" tasks over world model generations in a visually-rich game (Bleeding Edge).
Performance on these tasks are considered as a proxy metric / indicator of world model generation quality.

The paper conducts an extensive empirical case study on a visually-rich game (Bleeding Edge).
First, the authors investigate performance on ground truth data, validating alignment on valid frame and action sequences, and comparing to simple VLM baselines.
Then, the authors investigate design choices such as data composition for tuning the performance of the approach.
Lastly, the authors study the evaluation quality of the proposed method on the two tasks on data generated by world models by conducting a human study, where humans rate the outputs of the algorithm.

**Strengths:**

- The idea of using VLMs for semantic tasks evaluation as a proxy for generation quality evaluation is new and interesting.
- The method presents multiple options for training / fine-tuning different subsets of parameters. Notably, only updating the projection head (0.07% of params according to the paper) proves highly effective. This piece of evidence could be important for the community I believe.
- The paper is overall well-written, easy to follow, and thorough. The appendix is detailed and includes further results and analysis to support the method and the design choices.
- For the included experiments, the methodology is generally solid.
- Open source code, evaluation data, and human annotation data.

**Weaknesses:**

`W1`: The paper only presents VLM baselines for world model evaluation, which, according to the authors, were never used before for that purpose. How the method compares to simple world model evaluation approaches such as measuring mean squared error or FVD (between generations and ground truth using a prefix as initial context) using a held out test set, or training control policies under various reward signals (goals) through world model interaction and measuring success via cumulative returns? The paper should include a comparison to existing evaluation baselines.

`W2`: The results in Fig. 9 suggest poor accuracy of the method on setting 8, which involves data generated by a smaller model. The authors attributed this to a "resolution mismatch". However, I suspect that since smaller models tend to generate trajectories that deviate more significantly from the ground truth, the results could indicate that the method is perhaps more indicative for correct or higher quality generations, while it may be failing to identify cases where the world model generations diverged from the ground truth, which are the more important subset of cases to capture.

This is also related to the rollout filtering concern below.


`W3`: Rollout Filtering: I am concerned that the filtering procedure in the paper may filter out poor generation cases, leaving mostly high-quality, simpler (and easier to classify), rollouts. Importantly, in cases where the world model failed to generate the correct dynamics given the sequence of actions (conditioning), the output could be among the filtered sequences. Thus, the results may be indicative for successful generations, but not for poor ones.

Given the significant efforts put into this paper, I would suggest to collect a reasonable test set of cases where the generated content (dynamics) deviates from *the ground truth*, i.e., the actual dynamics under the same "raw" actions (for conditioning) starting from the same context, in a deterministic environment. Then, demonstrate that the proposed approach reliably captures these cases.
Preferably, also consider inaccurate world models such as the small model used in the paper. Aim for generating both (1) poor reconstructions examples (blur, artifacts, etc.), and (2) incorrect but convincing dynamics that look valid but deviate from the ground truth.


`W4`: The proposed evaluator outputs are given in natural language. Given only the description in the paper, it is unclear how this method would be applied in a practical world model evaluation setting (e.g., in a new environment). Specifically:
1. Does the method requires to collect some test set of trajectories from the real environment serving as reference trajectories?
2. How would the questions and answers automatically adapt to a new environment?
3. Suppose that we have generated a set of trajectories (inference) with a world model. The proposed evaluator emitted a corresponding set of answers / strings. How the approach determines the quality of the generations? (suppose each generated trajectory could be long and span multiple 14-frame segments). How performance is aggregated over a single trajectory that spans multiple segments?
4. Is the approach valid only for evaluating very short trajectory segments?
5. What exactly is required for applying this method in an independent application? and how to obtain a single aggregated score that reflects the generation quality?

The paper should includes the answers to these questions clearly.


`W5`: The method seems to rely heavily on the "Description Generation" step (line 188), for converting sequences of frames and raw actions to language form. It is non-trivial to assume that such an approach would work reliably in general domains. Here, inferring character movement direction in third person view could be much easier than intricate finger movements in first person view for example.

It is unclear how reliable this step is, i.e., how reliably it captures cases where the actions are not aligned with the generated frames (which typically occurs when the world model is given an action sequence that leads to a novel dynamics behavior that was not sufficiently explored / observed and thus is not properly represented in the training data). I suggest to carry a specific study on this aspect, similarly to that I proposed in `W3` above, and show that (hopefully) the proposed approach is indeed reliable.



`W6`: Claiming for effective world model quality evaluation method, as the paper title and abstract suggest, requires several diverse benchmarks (environments / games) with diverse dynamics (between benchmarks), studies that show evaluation quality over longer horizons (could be aggregated scores over 14 frame segments in your case), and more extensive studies, as suggested above. Demonstrating that a method works on a single game (although with multiple "stages" / "maps") does not sufficiently support such a general claim.


`W7`: While the motivation and claims suggest a very general method for world model evaluation, e.g., "a method for adapting VLMs to rollout evaluation under data and compute constraints" (abstract) and "establishing UNIVERSE as a lightweight, adaptable, and semantics-aware evaluator for world models." (abstract), in practice the approach is only valid for video environments, and only evaluates action and character recognition over very short horizons. I expect that the claims would represent the method more accurately, i.e., limit the scope of the claims.



`W8`: "Assesses whether generated sequences accurately reflect the effects of agent actions ***at each timestep***" (line 173).
This claim is inaccurate, as the method was only validated at a coarser action granularity (language description).


`W9`: line 1454, the variable letter `R` was abused, re-defined for both recall and the set of response n-grams.


`W10`: line 1768: large-scale and small scale models swapped?


`W11`:
> The task distribution favors Action Recognition (αAR = 0.8) ***due to its stronger causal grounding*** (line 255)

Please support this claim, it is non-trivial.


`W12`: The binary vs. multiple-choice vs. open-ended comparisons are presented throughout the results. First, it is not clear enough that these are complementary sets of QA (the comparison could also suggest that you compare different formats of presenting the Q/A). Second, while presenting the results for all QA types provide empirical insight, I think it is not the most important aspect of the work and thus it is somewhat confusing as it takes a very significant real-estate in the main paper. I believe that the result would be clearer by presenting aggregated scores first, to indicate overall performance clearly, in the same way that the final generation quality evaluation (score) is given, and present the results for the separate QA types later (or in the appendix).
To be clear, I do not consider this as a major weakness.

**Questions:**

`Q1`: in "Rollouts Generation" (line 1774), how exactly the action sequence used for conditioning is determined? How the 1 sec context was chosen?

`Q2`: "d = Describe(c(1:L), m)." (line 1223). How exactly `Describe` works? Is it trained specifically on data of this game? Is it necessary to train such model in order to use the method to evaluate world models in practice?

`Q3`: How the performance of UNIVERSE on character recognition in Figure 1 align with those in Figure 2 (right)? i.e., 99%+ vs. 84%?

---

> ### Author Response · Authors · 2025-11-21
> **Rebuttal by Authors 1/2**
>
> Thank you for your valuable review and helpful suggestions. We especially appreciated your suggestions on rollout filtering and clarification of description generation. We respond to your questions and concerns below.
>
> **Additional baselines**
>
> This is a fair and very reasonable point. We would like to clarify that we used VLM models here because we focus on the semantic evaluation of rollouts that require reasoning across visual and textual information. To this end, we used *several VLM baselines* (Figure 2) and we also tried all variants of *CLIPScore* [1], a standard vision-language evaluation metric (Appendix G3). In addition, we run a preliminary evaluation of *GPT-5* on the simplest protocol (Appendix G.1). All these methods underperformed.
>
> That said, we agree that comparisons with such baseline metrics as FVD [2] would be valuable. We added Appendix H, where we report raw FVD scores across all environments and then analyze their relationship to human-aligned UNIVERSE scores via Pearson correlation on the same set of rollouts (Appendix H, Table 18). The results show weak and inconsistent correlations, indicating that UNIVERSE captures complementary evaluative signals beyond FVD.
>
> In addition, we added a comparison text-to-video evaluation suite, vbench [3]. The analysis covered 8 environments, 6 evaluation settings, and 7 evaluation dimensions. We compared UNIVERSE human-aligned scores with those from VBench  on the same set of rollouts (Appendix H; Fig. 17). Similarly to FVD, the results show weak correlation, indicating that UNIVERSE complements VBench, too.
>
> **Results in Figure 9 (Setting 8)**
>
> We would like to clarify why we think the drop in Setting 8 is due to the resolution mismatch: UNIVERSE is pretrained at 224px, while the smallest world model used in Setting 8 outputs 128×128 rollouts. This mismatch affects both visual detail and the reliability of frame-level semantics. Notably, human annotators also struggled in this setting: they achieved their lowest accuracy (54.46%% on average, Figure 9), while the inter-annotator agreement remained high (κ = 0.79, Figure 9). Besides, in Setting 8, annotators marked 6/30 rollouts as unclear, compared to only 7 rollouts marked as unclear across all Settings 1-7. (see Table 14). The evidence suggests that the performance gap is due to the resolution difference.
>
> **Clarification on Rollout Filtering**
>
> Thank you for raising this important point. We would like to clarify that our filtering procedure was conservative and performed by a separate set of annotators, who were instructed to discard rollouts under a narrow set of mechanical criteria such as (i) early, uninformative segments, (ii) rollouts dominated by full-frame occlusion, and (iii) sequences where no agent was visible.
>
> Consequently, the filtering step does not remove “poor” or low-quality generations. In fact, many retained rollouts contain notable artifacts and errors—annotators evaluating UNIVERSE reported issues such as (i) agent skin and appearance drift, (ii) missing or corrupted assets, and (iii) overexposed heads-up display elements. These imperfect rollouts remain in the evaluation set, and UNIVERSE is required to handle and score them.
>
> To further clarify this point, we expanded Appendix F and added Figures 12 and 13 illustrating (i) examples excluded by the filtering step, and (ii) challenging rollouts retained for human evaluation, respectively.
>
> **Applying the method to new domains**
>
> We appreciate the request for clarification and have added a dedicated section to Appendix C.1. In summary, adapting UNIVERSE to a new environment would require only a small set of reference trajectories containing ground-truth observations/actions for the evaluation dimensions of interest. These trajectories serve to instantiate the Q/A templates and construct the mixed-supervision adaptation dataset.
>
> At inference time, UNIVERSE evaluates rollouts by emitting dimension-wise natural-language responses, which can either be used directly as structured feedback for improving the world model or mapped to numerical scores and aggregated into a single trajectory-level metric. Although we used 14-frame inputs in our experiments, this is a design choice rather than a limitation—the evaluator can be applied to longer rollouts by increasing the context window or using a sliding-window approach.
>
> Overall, deploying the evaluator in a new domain would require: (i) a small adaptation dataset of reference trajectories with ground-truth labels, (ii) the task specification, and (iii) the lightweight fine-tuning recipe.

---

> > ### Author Response · Authors · 2025-11-21
> > **Rebuttal by Authors 2/2**
> >
> > **Description Generation clarification and model generalization**
> >
> > We agree that the description-generation component is an important part of this work. In our study, descriptions were generated from controller logs and environment metadata, which provide precise ground-truth information about actions and states. We have clarified this in Appendix D.1.
> >
> > Regarding model generalization: the evaluator is designed to detect semantic misalignment between observations and world model input, even when the world model produces novel behaviors. Our human evaluation study demonstrates this capability: the world models we evaluated produced behaviors outside their training distribution (Fig. 13), and the evaluator was able to successfully evaluate the rollouts along the specified dimensions.
> >
> > Finally, if we want to adapt the evaluator to target behaviors that are rare or underrepresented in the dataset, our mixed-supervision strategy offers an opportunity to mitigate the imbalance by increasing the relative proportion of target cases in the data mix.
> >
> > **Paper Claims Clarification**
> > We agree that our claims should more clearly reflect the demonstrated scope. Our contribution is a low-cost adaptation method for evaluating video world models, validated on seven diverse virtual environments (plus a low-resolution variant) and two resolutions. While the environments differ meaningfully in dynamics, we acknowledge that they remain within the virtual/video domain. Extending to additional domains and longer horizons is an important direction for future work and is now stated explicitly in the Limitations. In addition, we revised the paper title, abstract, and introduction to clarify the claims.
> >
> >
> > **The binary vs. multiple-choice vs. open-ended comparisons**
> >
> > Our intention in reporting all three QA types was to analyze evaluator performance at different levels of semantic complexity. This granularity proved informative, for example, during preliminary evaluation of GPT-5 on the simplest protocol evaluation setting (see Appendix G.1).
> > Furthermore, QA-type stratified evaluation supports our goal of developing a single unified evaluator that can perform on par with six task-specific checkpoints while using less data and training time (Fig. 1). That said, we agree that we should clarify that these QA are complementary. We now clarify this explicitly in the Method section.
> >
> > We also note that, for most of the main results (Fig. 2, 3, 4, 7, 9), we already present dimension-level aggregated scores, reserving per-QA-type breakdowns for more detailed analysis. We retain the unaggregated view in Fig. 1 only to provide a deeper initial illustration.
> >
> > **Rollouts Generation clarification (line 1774)**
> >
> > For rollout generation, we use reference trajectories from the game dataset that include time-aligned controller inputs and image frames. For each rollout, we randomly sample a 1-second context window (video frames and corresponding action tokens) from these trajectories. The world model is then conditioned on this context, which consists of an interleaved sequence of tokenized images and actions, and proceeds autoregressively to generate latent image tokens and discretized action tokens. We update Appendix F.1 to clarify it.
> >
> >
> > **"d = Describe(c(1:L), m)" clarification (line 1223)**
> >
> > `Describe` is a deterministic procedure that uses the controller logs and environment metadata available in our reference trajectories to extract the relevant ground-truth information. It parses these logs and extracts relevant information, which is then used to construct the QA pairs. We updated Appendix D.1 to clarify it.
> >
> > **Performance of UNIVERSE in Figure 1 vs. Figure 2**
> >
> > The discrepancy arises because the figures use different metrics and training regimes: Fig 1 reports performance after 10 epochs and reports EM, whereas Fig 2 reports results after 1 epoch using ROUGE. We have refined both caption to clarify it.
> >
> > **Wording issues and typos**
> >
> > We thank the reviewer for pointing these out. We improved the phrasing and fixed typos:
> >   - W8: We revised the description to state that we focus on semantic alignment given the selected segment.
> >   - W9-10: typos fixed.
> >   - W11: rephrased the sentence to emphasize AR speed of convergence.
> >
> > ---
> > [1] Hessel, Jack, Ari Holtzman, Maxwell Forbes, Ronan Le Bras, and Yejin Choi. "Clipscore: A reference-free evaluation metric for image captioning." In Proceedings of the 2021 conference on empirical methods in natural language processing, pp. 7514-7528. 2021.
> >
> > [2] Unterthiner, T., Van Steenkiste, S., Kurach, K., Marinier, R., Michalski, M., & Gelly, S. (2018). Towards accurate generative models of video: A new metric & challenges. arXiv preprint arXiv:1812.01717.
> >
> > [3] Huang, Z., He, Y., Yu, J., Zhang, F., Si, C., Jiang, Y., ... & Liu, Z. (2024). VBench: Comprehensive benchmark suite for video generative models. In Proceedings of the IEEE/CVF Conference on Computer Vision and Pattern Recognition (pp. 21807-21818).

---

> > > ### Comment · Reviewer_dQyk · 2025-11-27
> > >
> > > I appreciate the authors’ thoughtful and detailed response. Several of my earlier concerns, particularly regarding rollout filtering and the performance discrepancy in setting 8, were effectively addressed, and the added comparison with FVD provides concrete evidence of improvement over a semantic-agnostic baseline such as FVD under the presented setup.
> > >
> > > However, after a deeper rereading of the paper during the rebuttal period, I identified additional concerns, one of which I believe is critical and remains unresolved. These are detailed below.
> > >
> > > #### Action-Agnostic Evaluator
> > >
> > > If I understand correctly, the VLM evaluator depends only on the sequence of frames (and the question), and is independent of the actions taken? Suppose a world model were to overfit a small set of trajectories and, at inference time, simply replay one of these trajectories regardless of the provided actions. How would the evaluator detect such failure cases?
> > >
> > > #### Evaluation on Additional Environments
> > >
> > > I still believe that demonstrating UNIVERSE on an additional game would significantly strengthen the claims. The experimental setup and the method involve many intricate details. It is not entirely clear to what extent the method would generalize to new and interesting environments.
> > >
> > > For example, in the demonstrated game the character is almost always visible, and questions depend only on that character and its relative movement.
> > >
> > > 1. This may explain the reported generalization: background details are irrelevant, and the model may simply learn to ignore them.
> > > 2. In realistic domains (e.g., robotics), identifying when a question applies is itself nontrivial (e.g., “Is the drawer open?” only matters when the drawer is visible).
> > >
> > > The takeaway is that it is unclear to what extent the method is realistically extendible to new environments and questions while retaining solid performance. It is possible that additional challenges would arise. In that sense, the current evidence is not convincing enough for supporting a claim such as a (general) evaluator of video world models.
> > >
> > > That said, I do appreciate that the current version demonstrates the viability of the method on such relatively simple setup.
> > >
> > > At its current stage, I believe the work would be stronger if framed as a **case study** highlighting a promising direction and inviting further exploration of its generality, rather than as an already-established general-purpose evaluation tool of video world models across diverse environments.
> > >
> > > #### The Costs Involved
> > >
> > > In a realistic usage of the method, I assume it is more likely that manual trajectory labeling would be necessary, as modifying the environment to include the desired information is not always possible (usually it is either not viable or not possible). Manually labeling even a small adaptation dataset is a costly task. The lack of an additional benchmark (game) is perhaps indicative that applying the method to a new environment is challenging. While community efforts could eventually mitigate this through shared annotated datasets, I still consider this a (relatively minor) practical limitation, compared to existing evaluation methods.
> > >
> > >
> > >
> > > #### Lack of Numerical Score
> > >
> > > By default, the current evaluation yields natural-language outputs rather than a numerical metric, which requires further work to be mapped to a score in practical settings. The complexity of this work perhaps depends on the specific evaluation and setting. As an evaluation tool, the ability to provide a score is an important practical consideration.
> > >
> > >
> > >
> > > #### Limitations of Discrete Evaluation Metrics
> > >
> > > The evaluation method is inherently discrete and lacks sensitivity to continuous variation. For example, two trajectories may both be classified as either success or failure, while the differences in their relative quality would not be reflected in the evaluation. While this may not be a significant limitation, as the method can be used in conjunction with other evaluation methods, it is worth mentioning.
> > >
> > > -----
> > >
> > > Thank you again for the clear and constructive response. The concern regarding the evaluator’s independence from actions, which I noticed during a deeper rereading, remains unresolved and, if correct, poses a critical limitation that substantially weakens the central claims. Since this issue emerged late in the rebuttal period, I understand the authors could not fully address it, but clarification is still necessary.
> > >
> > > I am keeping my score unchanged for now, but I am open to increasing it if the authors can address this point convincingly.

---

> ### Author Response · Authors · 2025-11-29
> **Rebuttal by Authors 1/2**
>
> Thank you for your thoughtful and constructive follow-up. We sincerely appreciate the time and effort you have dedicated to this discussion, your detailed feedback has significantly strengthened our work. Below, we address each concern you raised:
>
>
> **Action-Agnostic Evaluator**
>
> Thanks for this question, it addresses an important aspect of our evaluation framework that requires further clarification. At inference time, UNIVERSE supports two evaluation modes: (i) ground-truth–aligned evaluation, and (ii) open-ended evaluation. The failure case you describe (where a world model ignores the conditioning actions and replays memorized trajectories) falls under (i).
>
> In this setting, we use the same action sequence that was used to condition the world model to instantiate QA pairs with known correct answers. UNIVERSE then receives both the generated frames and the corresponding action-conditioned question. Because the ground-truth answer (input action) is known, any mismatch is directly exposed.
>
> To make this clearer, we have added Appendix C.3 to elaborate on this in more detail.
>
> **Evaluation on Additional Environments**
>
> We acknowledge that our experimental validation focuses on visually-rich game environments where the agent is typically either fully or partially visible. Adapting UNIVERSE to domains with different visual characteristics, such as robotics scenarios, would require domain-specific design choices.
>
> For example, in the robotics scenario that you mentioned ("Is the drawer open?"), a structured evaluation approach would involve: (1) identifying objects of interest, (2) determining when the target object is visible, and (3) querying object properties (drawer state). This hierarchical evaluation strategy is conceptually compatible with our framework, but would require careful adaptation as described in Appendix C.2
>
> That said, we agree that the current submission might be better framed as a case study demonstrating the viability of lightweight VLM adaptation for semantic world-model evaluation.
>
> We have revised the paper accordingly.
>
> **The Costs Involved**
>
> We agree that data acquisition costs are an important factor when deploying models in new domains.
>
> Adapting UNIVERSE to a novel domain may require manual labeling when ground-truth metadata (such as controller logs or environment state) is unavailable. However, we emphasize that our method is designed to minimize this cost: once a small adaptation dataset is obtained, even with manual annotation, the evaluator can be efficiently fine-tuned using our lightweight adaptation recipe. This stands in contrast to training task-specific models from scratch or relying on expensive human evaluation for every rollout assessment.
>
> Our focus on virtual environments is motivated by their accessible ground-truth metadata, which allowed us to investigate the adaptation procedure without incurring human labeling costs. Extending to different virtual environments would require either (i) access to environment-side metadata (as done in this work) or (ii) collection of human-labeled trajectories. Both paths involve coordination and resource investment beyond the scope of this case study.
>
> That said, we view our current work as demonstrating a promising direction, with expansion to additional domains as valuable future work.
>
> We have revised Appendix C.2 to clarify this point.

---

> ### Author Response · Authors · 2025-11-29
> **Rebuttal by Authors 2/2**
>
> **Lack of Numerical Score**
>
> We deliberately chose natural-language outputs with the goal of providing interpretable, structured feedback; the long-term goal is to enable direct semantic feedback that can be passed to the language-conditioned world model during training. The diversity of QA formats (binary, multiple-choice, open-ended) allows UNIVERSE to express richer semantic judgments than a single scalar score.
>
> That said, we agree that numerical scores are practically important for benchmarking and comparison. UNIVERSE supports straightforward conversion to numerical metrics depending on the evaluation setup:
>
> (i) ground-truth–aligned evaluation: because correct answers are known, we can convert UNIVERSE’s predictions into numerical metrics using EM and ROUGE over ground truth and predicted answers;
> (ii) open-ended evaluation: here, no ground-truth answers exist, but UNIVERSE’s responses can be aggregated into meaningful statistics.
>
> We have added a discussion of this distinction to Appendix C.3.
>
>
> **Limitations of Discrete Evaluation Metrics**
>
> Thank you for highlighting this. While our QA protocol is discrete at the format level, the evaluator is, in fact, sensitive to graded semantic differences along the specified dimensions. For example, if two trajectories both involve an agent performing two different actions, UNIVERSE can distinguish whether the target action occured in each trajectory (binary QA), and which action occured (multiple-choice or open-ended QA).
>
> Thus, although individual QA items are discrete, the semantic resolution of the evaluator is not limited to binary success/failure. Open-ended responses in particular allow our method to express nuanced semantic distinctions across rollouts.
>
> ---
> We hope these clarifications and the revised manuscript address your concerns. We welcome any further questions or suggestions you may have.

---

### Official Review · Reviewer_SfSa · 2025-11-01

**Soundness:** 4
**Presentation:** 4
**Contribution:** 4
**Rating:** 8
**Confidence:** 3

**Summary:**

This paper presents a method for training a VLM to evaluate the action and character consistency of a world model. The main contributions are:

1) A large-scale gameplay dataset for training and evaluating world models.
2) A pipeline for generating inconsistent actions and characters from the ground truth data along with question-answer pairs.
3) A parameter-efficient method for training a VLM to evaluate the action and character consistency of a world model.

**Strengths:**

Overall this problem is very timely given recent interest in world models. While this work focuses on a single game, they extensively ablate the different components of their method and show that their approach is effective and likely generalizable to other environments. Particularly the authors thoroughly ablate the training data composition and VLM training method. Additionally, the authors provide a thorough evaluation of the model's generalization to out-of-distribution data by evaluating the model on held-out environments and rollouts from eight different world models.

**Weaknesses:**

The main weakness, as mentioned by the authors in the limitations section, is whether this method can be applied beyond video games to other environments. The training data construction method relies on action logs from the game which are not available for other environments and might be costly to acquire for a large set of environments. The problem is likely compounded for real-world simulators used for robotics and other embodied agents.

**Questions:**

1) How do you scale Universe to a wider set of gaming environments? Would this require partnerships with more game studios? (likely costly and limiting)
2) You mention that applying Universe to long-horizon trajectories is an open problem. Is this simply due to the training samples being 14-frames long? Or are the models less capable of reasoning across longer image sequences?

---

> ### Author Response · Authors · 2025-11-21
> **Rebuttal by Authors**
>
> Thank you for the thoughtful review and for highlighting several important considerations. We especially appreciated your point on the implications of scaling the proposed approach to novel domains. We address your points below.
>
>
> **Generalizations to other domains**
>
> We agree that extending UNIVERSE beyond video-game environments is an important direction for future work. We chose virtual environments for this initial study because they provide ground-truth state, controlled variability, and full reproducibility. Beyond practicality, simulated/game environments are increasingly valuable in their own right for AI testing, content generation, and player modeling [1].
>
> That said, we fully acknowledge that applying UNIVERSE to robotics or real-world simulators would require additional effort—primarily data collection, dataset construction, and domain-specific tuning. We have updated the Limitations section to make this scope explicit.
>
> When it comes to scaling our approach to a broader set of environments, we believe there are two primary practical routes one could follow:
>   * Partnerships with game studios (as in the present work), which provide high-quality logs and metadata but are resource-intensive.
>   * Bootstrapping from publicly available game video datasets, where generative models or synthetic data could be used to create initial supervision signals.
>
> We have added Appendix C.2 and discussed it there in more detail.
>
>
> **Applying UNIVERSE to long-horizon trajectories**
>
> Thank you for raising this point. Our current experiments focus on a length of up to 14 frames because of the granularity of the target tasks. That said, we hypothesize that our approach can be applied to longer-horizon tasks either by extending the visual context or by applying a sliding window approach. In addition, our frame-sampling analysis indicates that not all frames contribute equally to performance, suggesting that more intelligent sampling or hierarchical summarization could further extend UNIVERSE to longer horizons. Future work will explore improved frame-sampling strategies and context-scaling methods to enable robust long-horizon evaluation. We have updated our limitations section to clarify this.
>
> ---
> [1] Kanervisto A, Bignell D, Wen LY, Grayson M, Georgescu R, Valcarcel Macua S, Tan SZ, Rashid T, Pearce T, Cao Y, Lemkhenter A. World and human action models towards gameplay ideation. Nature. 2025 Feb 20;638(8051):656-63.

---

### Official Review · Reviewer_ED2d · 2025-11-03

**Soundness:** 2
**Presentation:** 3
**Contribution:** 2
**Rating:** 4
**Confidence:** 4

**Summary:**

This paper presents UNIVERSE (UNIfied Vision-language Evaluator for Rollouts in Simulated Environments), a framework for evaluating the quality of video rollouts generated by world models. The authors propose a structured evaluation protocol comprising two identification tasks: Action Recognition (AR), which assesses whether generated videos align with input action sequences at the timestamp level, and Character Recognition (CR), which evaluates entity identity consistency over time. Each task is evaluated through three question-answering formats (binary, multiple-choice, and open-ended). UNIVERSE is built upon the PaliGemma 2 3B architecture and adapts the model by fine-tuning only the multimodal projection head (0.07% of parameters). The authors conduct extensive experiments totaling 5,154 GPU-days and validate their approach through human evaluation across 8 environment settings.

**Strengths:**

1. **Well-motivated problem with practical importance:** The paper addresses a genuine need in the world model community for fine-grained, semantically-aware, and temporally-grounded evaluation methods. The limitations of low-level metrics like FID/FVD are well-articulated.

2. **Structured and actionable evaluation protocol:** The AR/CR task decomposition with multiple QA formats provides a clear, operationalizable framework that other researchers can adopt and extend. This structured approach is more systematic than ad-hoc human evaluation.

3. **Comprehensive experimental validation:** I appreciate the thoroughness of the ablation studies examining fine-tuning configurations, frame sampling strategies, supervision amounts, and data mixing ratios. These provide valuable empirical insights for practitioners.

4. **Rigorous human evaluation:** The human study design (240 rollouts, 1440 QA instances, multiple annotators with adjudication, Cohen's κ reporting) demonstrates careful attention to validation methodology.

5. **Resource efficiency:** Achieving competitive performance while fine-tuning only 0.07% of parameters is practically valuable for researchers with limited computational resources.

**Weaknesses:**

1. **Missing comparison with existing evaluation benchmarks:** My primary concern is that the paper motivates UNIVERSE by suggesting existing benchmarks lack certain capabilities, but I could not find any direct experimental comparison with methods like VBench or EvalCrafter on the same data. I would strongly encourage the authors to provide such comparisons, specifically measuring how different evaluation protocols correlate with human judgments on identical rollouts. This would substantiate the claim that UNIVERSE offers necessary improvements over existing tools.

2. **Unclear positioning of methodological contribution:** The core technical approach—fine-tuning only the projection head—appears to be a well-established practice in VLM training literature (often called the "alignment stage"). While applying this effectively to a new domain is valuable, I suggest the authors clarify whether this represents a novel discovery or an effective application of known techniques. This would help readers better understand the nature and scope of the contribution.

3. **Incomplete coverage of foundational related work:** I notice that the paper does not cite some foundational work in the "model-as-judge" paradigm, particularly Zheng et al. (2023) "Judging LLM-as-a-Judge with MT-Bench and Chatbot Arena," which pioneered using models as automatic evaluators. Since UNIVERSE directly extends this paradigm to multimodal evaluation, I recommend including this reference to provide clearer intellectual lineage.

4. **Limited scope of generalization validation:** While the paper evaluates 8 settings, all data comes from a single game (Bleeding Edge) environment. I observed that performance degrades noticeably at lower resolutions (Setting 8: 56.55% on AR), suggesting sensitivity to distribution shifts. I would encourage testing on more diverse world model types (e.g., autonomous driving, robotics) to better understand generalization capabilities.

5. **Potential circular reasoning in baseline comparisons:** The "task-specific baselines" mentioned in the paper appear to be other variants trained by the authors rather than established external methods. While these comparisons are informative for understanding different adaptation strategies, I suggest clearly distinguishing between internal architectural ablations and comparisons with established community benchmarks.

**Questions:**

1. **Comparison with existing benchmarks:** Could you provide experimental results comparing UNIVERSE's scores with those from VBench and EvalCrafter on the same set of rollouts? Specifically, I would be interested in seeing correlation coefficients (Spearman/Pearson) between each method's scores and your collected human judgments. This would help clarify how UNIVERSE's evaluation protocol relates to and potentially improves upon existing approaches.

2. **Methodological positioning:** Could you clarify whether the projection-head-only fine-tuning approach was explored as a known technique from VLM training literature, or whether you discovered it independently through your experiments? If it builds on known practices, how does your application differ or extend them?

3. **Generalization to other domains:** Have you conducted any preliminary experiments on world models from domains other than gaming (e.g., robotics, autonomous driving)? What challenges do you anticipate for adapting UNIVERSE to these domains?

4. **Data efficiency analysis:** Given the large computational investment (5,154 GPU-days), could you provide more insight into which experiments contributed most to your key findings? This would help the community understand resource allocation for similar adaptation tasks.

5. **Baseline clarification:** When you mention achieving "comparable performance to task-specific baselines," could you clarify which specific external methods or benchmarks these refer to, versus internal architectural variants?

---

> ### Author Response · Authors · 2025-11-21
> **Rebuttal by Authors**
>
> Thank you for your constructive review and feedback. We especially appreciated your suggestion on investigating the relationship between our methods and existing text-to-video benchmarks! Below, we provide clarifications to your questions and concerns.
>
>
> **Comparison with existing evaluation benchmarks**
>
> This is a very helpful and fair request. We added a direct correlation study between scores of UNIVERSE and VBench on the same set of rollouts. The analysis covers 8 settings, 6 protocol dimensions (2 tasks × 3 prompt types), and 7 evaluation aspects. In addition, we compared UNIVERSE scores with FVD computed between rollouts and ground truth on the same set of rollouts. Details, including numeric tables and plots, are included in the revised paper (Appendix H; Fig. 17; Table 18). The high-level result indicates that UNIVERSE complements existing both VBench and FVD, which we attribute to its ability to capture fine-grained semantic alignment.
>
>
> **Clarification of methodological contribution**
>
> While part of our approach focuses on partial fine-tuning (tuning the projection head), we would like to clarify that our contribution is a practical, low-cost adaptation recipe that turns a general VLM into a unified evaluator for world-model rollouts. Specifically, the recipe combines projection-head tuning, efficient frame sampling, and mixed supervision, enabling reliable alignment with human judgments across diverse environments at minimal cost. Hence, the novelty lies in the integration and application of these elements to the world-model evaluation setting. We revised the Introduction to make it clearer.
>
> **Performance at lower resolutions (setting 8)**
>
> We would like to clarify why we think the drop in Setting 8 is due to the resolution mismatch: UNIVERSE is pretrained at 224px, while the smallest world model used in Setting 8 outputs 128×128 rollouts. This mismatch affects both visual detail and the reliability of frame-level semantics. Notably, human annotators also struggled in this setting: they achieved their lowest accuracy (54.46% on average, Figure 9), while the inter-annotator agreement remained high (κ = 0.79, Figure 9). Besides, in Setting 8, annotators marked 6/30 rollouts as unclear, compared to only 7 rollouts marked as unclear across all Settings 1-7. (see Table 14). The evidence suggests that the performance gap is due to the resolution difference.
>
> **Generalization to novel domains**
>
> We chose virtual environments for this initial study because they offer ground-truth state, controlled variability, and full reproducibility. Beyond practicality, simulated/game environments are increasingly valuable in their own right for AI testing, content generation, and player modeling [1]. We revised our introduction and conclusion to clarify them.
>
> That said, we agree that extending our approach to more domains is an exciting direction for future work. Deploying the evaluator in a new domain would require: (i) a small adaptation dataset of reference trajectories with ground-truth labels, (ii) the task specification, and (iii) the lightweight fine-tuning recipe described in the paper. We added Appendix C.1 to discuss it in more detail.
>
> **Terminology clarification**
>
> We agree that the “task-specific baselines” wording was misleading. Our “task-specific baselines” are internal checkpoints, not external community benchmarks. Their purpose is to demonstrate that mixed supervision and efficient frame sampling allow a single unified evaluator to match or exceed the performance of individually trained task-specific checkpoints while being trained only 1/6 of the time. We revised the text to clarify this distinction.
>
> When it comes to external baselines, we compare UNIVERSE to five external baselines (Fig. 2) and run ablations that motivate our design choices (Section 5).
>
> **Data efficiency analysis**
>
> We clarify that the reported 5,154 GPU-days reflects the entire study, not the cost of training UNIVERSE. The adaptation step itself is lightweight. Most compute was spent on baseline training and large-scale analysis: analysis and ablations (2,554 GPU-days), baseline fine-tuning (864 GPU-days), zero-shot evaluations (136 GPU-days), and human evaluation (~1 GPU-day).
>
> **Additional related work** Thank you for pointing it out, we added [2] to the paper!
>
> ---
> [1] Kanervisto A, Bignell D, Wen LY, Grayson M, Georgescu R, Valcarcel Macua S, Tan SZ, Rashid T, Pearce T, Cao Y, Lemkhenter A. World and human action models towards gameplay ideation. Nature. 2025 Feb 20;638(8051):656-63.
>
> [2] Zheng, L., Chiang, W. L., Sheng, Y., Zhuang, S., Wu, Z., Zhuang, Y., ... & Stoica, I. (2023). Judging llm-as-a-judge with mt-bench and chatbot arena. Advances in neural information processing systems, 36, 46595-46623.

---

### Author Response · Authors · 2025-11-21
**Rebuttal by Authors**

We thank all the reviewers for taking the time to review our paper and for their constructive and valuable feedback.

We are encouraged that reviewers recognize our work as addressing a well-motivated and timely problem (ED2d, SfS) with a novel and interesting approach (dQyk), supported by comprehensive methodology and evaluation (ED2d, SfS, dQyk) and rigorous human validation (ED2d). We appreciate the recognition of our practical contributions, including the resource-efficient adaptation strategy (ED2d, dQyk), structured evaluation protocol that others can adopt (ED2d), and our commitment to open science through releasing code, data, and annotations (dQyk).

In response to the reviewers’ feedback, we made substantial revisions to both the main paper and the appendix:
  * Improved Evaluation and Baselines:
      * Added comparison with FVD [1] on identical rollout sets (Appendix H.1, Table 18; reviewer dQyk)
      * Added comparison with VBench [2] metrics on identical rollout sets (Appendix H.2, Figure 17; reviewer ED2d)
  * Improved Clarity and Methodology:
      * Revised Introduction to better position our methodological contributions (reviewer ED2d)
      * Clarified causal grounding claim for Action Recognition task weighting (Section 3, lines 260-261; reviewer dQyk)
      * Improved description of rollout filtering and added examples of excluded (Figure 12) and retained challenging cases (Figure 13) (Appendix F; reviewer dQyk)
      * Clarified description generation process (Appendix D.1; reviewer dQyk)
      * Discussed details on which experiments contributed most to our findings (reviewer ED2d)
      * Clarified rationale for evaluating across multiple prompt types (reviewer dQyk)
      * Discussed rollouts generation process in more detail (Appendix F.1; reviewer dQyk)
      * Refined captions of Figures 1 and 2 to clarify UNIVERSE performance (Appendix D.1; reviewer dQyk)

  * Generalization and Scope:
      * Clarified model performance degradation at lower resolution in Setting 8 (reviewers ED2d, dQyk)
      * Enhanced discussion of generalization to novel domains (Appendix F.1; reviewers ED2d, SfS, dQyk)
      * Expanded Conclusion with guidance on applying UNIVERSE to long-horizon trajectories (reviewer SfS)
  * Related Work and Polish:
      * Added missing related work citations (reviewer ED2d)
      * Improved wording throughout (reviewers ED2d, SfS)
      * Fixed typos and formatting issues (reviewer SfS)

We believe these revisions substantially strengthen the paper and address all major concerns. We remain available to discuss any remaining questions and look forward to continued dialogue with the reviewers.

---
[1] Unterthiner, T., Van Steenkiste, S., Kurach, K., Marinier, R., Michalski, M., & Gelly, S. (2018). Towards accurate generative models of video: A new metric & challenges. arXiv preprint arXiv:1812.01717.

[2] Huang, Z., He, Y., Yu, J., Zhang, F., Si, C., Jiang, Y., ... & Liu, Z. (2024). Vbench: Comprehensive benchmark suite for video generative models. In Proceedings of the IEEE/CVF Conference on Computer Vision and Pattern Recognition (pp. 21807-21818).

---

### Author Response · Authors · 2025-12-03
**Rebuttal by Authors Part 2**

We would like to thank the reviewers once again for their constructive feedback, as well as the area chairs and organizers for their efforts in managing the recent incident.

As outlined in our earlier message, we addressed all initial reviewer concerns and updated the manuscript accordingly. We would also like to note for the new Area Chair that reviewer dQyk raised several follow-up points during the discussion period and stated that clarification of the Action-Agnostic Evaluator was the primary factor for increasing their score. We have now provided a detailed clarification resolving this main concern, along with responses to their other points. Due to the discussion freeze, the reviewer is unfortunately unable to acknowledge these updates or adjust their score.

To address the additional concerns of reviewer dQyk, we further revised the paper as follows:
* Clarified that our evaluator is not action-agnostic and added Appendix C.3 with a detailed explanation.
* Emphasized more clearly in the submission that the work is presented as a case study.
* Discussed adaptation costs in Appendix C.2.
* Discussed natural-language vs. numerical scoring in Appendix C.3.
* Addressed the semantic sensitivity of our evaluator.

We respectfully highlight that all reviewer concerns have been fully addressed, and we hope this context is helpful for the Area Chair’s evaluation.

---

### Meta-Review · Area_Chair_oVvc · 2026-01-06

**Summary:**

The paper received divergent evaluations （4、8、2）.

Reviewer ED2d (4): Questions the methodological positioning and novelty, noting the lack of direct comparison with established world-model evaluation benchmarks (e.g., VBench, EvalCrafter), limited generalization evidence beyond a single game environment, and potential circularity in baseline comparisons.

Reviewer SfSa (8，confidence：3): Largely supportive, also raises concerns about scalability beyond video games, given reliance on action logs, and highlights open questions regarding evaluation of long-horizon trajectories.

Reviewer dQyk (2): Raises fundamental concerns about evaluation validity, including missing comparisons to traditional world-model metrics, possible bias introduced by rollout filtering, limited ability to detect low-quality or divergent generations, unclear applicability beyond short segments, and over-claimed generality relative to the experimental evidence.

**Reviewer Concerns:**

Regarding the concerns, the authors provided detailed responses in the rebuttal.

Reviewer ED2d and Reviewer SfSa did not offer further feedback after the rebuttal, while Reviewer dQyk indicated that several key concerns remain unaddressed, including the action-agnostic nature of the evaluator, the computational costs involved, the lack of a clear numerical scoring mechanism, and the limitations of using discrete evaluation metrics.

**Reviewer Scores:**

Considering that Reviewer SfSa is supportive but expresses relatively low confidence, I place greater weight on the concerns raised by Reviewer ED2d and Reviewer dQyk. As these concerns remain substantial, I therefore recommend rejection.

---

### Decision · Program_Chairs · 2026-01-26

Reject